

# Retrieval of the land-sea contrast of cloud liquid water path by applying a physical inversion algorithm to combined zenith and off-zenith ground-based microwave measurements

Vladimir S. Kostsov[1], Dmitry V. Ionov[1], Anke Kniffka[2]

[1] Department of Atmospheric Physics, Faculty of Physics, St. Petersburg State University, Russia

[2] Zentrum für Medizin-Meteorologische Forschung, Deutscher Wetterdienst, Freiburg, Germany

*Correspondence to:* Vladimir S. Kostsov (v.kostsov@spbu.ru)

**Abstract.** Combined zenith and off-zenith ground-based observations by modern microwave radiometers provide an opportunity to study horizontal inhomogeneities of the humidity field in the troposphere and of the cloud liquid water path
(LWP) spatial distribution. However, practical applications are difficult and require thorough analysis of the information content of measurements, assessment of errors of data processing algorithm and the development of the quality control procedures. In this study we analyse the application of our LWP retrieval algorithm based on the inversion of the radiative transfer equation to the problem of detection of the LWP horizontal inhomogeneities by means of ground-based microwave observations in the vicinity of a coastline of a water object of medium size. The study is based on data acquired by the
microwave radiometer RPG-HATPRO which is located in the suburbs of St.Petersburg, Russia, at 2.5 km distance from the coastline of the Neva Bay (the Gulf of Finland) and is operating in angular scanning mode in the vertical plane. The retrieval setup is organised in such a way that zenith and off-zenith measurements provide equal sensitivity to atmospheric parameters. The optimal elevation angles for off-zenith observations are selected. The possibility to detect LWP horizontal inhomogeneity, namely the LWP land-sea contrast, for different measurement geometries (elevation angles) and values of
cloud base height is analysed. It is shown that ground-based microwave observations in the vicinity of a coastline can be a valuable tool for validation of the space-borne measurements of the LWP land-sea contrast if three principal requirements are met: (a) the multi-parameter physical inversion method is used for retrieving LWP; (b) rigorous bias correction and quality control procedures are applied to the retrieval results; (c) the information on the cloud base height is available. As a result of processing the microwave measurements at the observational site of St.Petersburg State University, the monthly-
averaged values of the LWP land-sea difference have been obtained for summer months within the period 2013-2021. For 24 out of 25 months of high quality observations, the LWP land-sea monthly difference is positive (larger values over land and smaller values over water) and can reach 0.06-0.07 kg m$^{-2}$. The estimations of the LWP land-sea contrast obtained from the ground-based microwave measurements at the observational site of St.Petersburg University are in very good agreement with the values of the LWP land-sea contrast obtained from the multi-year space-borne measurements by the SEVIRI
instrument (Spinning Enhanced Visible and InfraRed Imager) in the region of the Neva Bay (the Gulf of Finland) in June



and July. For August, the so-called "August anomaly" detected by space-borne observations is not confirmed by the ground-based measurements.

**Keywords:** cloud liquid water path; remote sensing; inversion algorithm; ground-based microwave radiometer; RPG-HATPRO instrument; horizontal inhomogeneities of atmospheric parameters

# 1 Introduction

## 1.1 Background

Passive ground-based microwave (MW) radiometry provides a possibility to detect horizontal inhomogeneities of atmospheric parameters relevant to air humidity and clouds and to plot maps of their spatial distributions if observations are performed in angular scanning mode (elevation scans and azimuth scans). There are several radiometers specially designed for that purpose. Crewell et al., (2001) presented the 22-channel radiometer MICCY (MIcrowave Radiometer for Cloud CarthographY). This radiometer is characterised by high temporal (1 s) and spatial (antenna beam 1°) resolution of measurements. Scanning can be done in horizontal (0-360°) and vertical (0-90°) planes and can be used for mapping clouds. This radiometer is transportable and therefore is suitable for mobile measurements. Another instrument for tropospheric monitoring is ASMUWARA (the All-Sky MUlti WAvelength Radiometer). It is a 10-channel system designed to observe the sky in all directions with an angular resolution of 9° (Martin et al., 2006a). The purpose of this instrument is to create maps of integrated water vapour (IWV) and cloud liquid water path (LWP). Several examples of such maps can be found at http://www.iapmw.unibe.ch/research/projects/ASMUWARA/online/, last access: 15 May, 2019. Martin et al. (2006b) presented the LWP retrieval algorithm used for processing microwave measurements by ASMUWARA. The paper also contains LWP sky maps and corresponding photographs of the sky. The good potential of a full-scanning 14-channel MW radiometer RPG-HATPRO (Radiometer Physics Gmbh – Humidity And Temperature PROfiler) for detecting horizontal water-vapour variability was demonstrated by Schween et al. (2011). It was shown that strength and direction of the air humidity horizontal gradient can be derived with a temporal resolution of about 15–20 min. However, to achieve this goal, the application of a simple linear-gradient model together with an assumed humidity vertical profile was necessary. The RPG-HATPRO instrument was also used to study the influence of a heterogeneous land surface on the spatial distribution of atmospheric water vapour (Marke et al., 2020): the measurements of integrated water vapour were performed during clear sky conditions at 30° elevation angle (full azimuth scans with 10° step). One more microwave instrument should be mentioned - SPIRA (Scanning Polarimetric Imaging RAdiometer) which operates at the frequency of 91 GHz and continuously scans the sky over a range of elevation angles (Stahli et al., 2011). A brief overview of observations of cloud liquid water by different ground-based MW scanning radiometers is given by Westwater et al. (2004).



In the 1980s, the tomographic approach to the retrieval of LWP was proposed. This approach is based on the use of moving platforms (air-borne and ground-based) for MW observations with angular scanning. Later, Huang et al. (2010) analysed the feasibility of this method for derivation of the spatial structure of cloud liquid water from measurements by modern microwave instruments. The application of a tomographic approach to retrieving two-dimensional water vapour fields was investigated by Meunier et al. (2015) by means of simulated experiments. The goal of the mentioned study was to

show how the instrument setup (number and spacing of elevation angles of instruments, number of frequencies, etc.) affected the quality of the retrievals.

    Previously, the measurements of LWP by the SEVIRI and AVHRR satellite instruments demonstrated the evidences of the systematic difference between the cloud amount and the LWP values over land and over water areas in Northern Europe (Karlsson, 2003; Kostsov et al., 2018b, 2019, 2021). Kostsov et al. (2020) have made an attempt to detect such

differences by means of ground-based microwave observations performed near the coastline of the Gulf of Finland in the vicinity of St.Petersburg, Russia. For this purpose, the radiometer RPG-HATPRO has been used which is located 2.5 km from the coastline and is operating in the angular scanning mode with the line-of-sight oriented in the direction of the coastline. In this way, the radiometer is probing the air portions over land (at elevation angle 90°) and over water area (at 7 elevation angles in the range 4.8°-30°). The statistical linear regression method was applied to the microwave measurements

by RPG-HATPRO in two spectral channels 31.4 GHz and 22.24 GHz. The LWP land-sea contrast was defined as the difference between the LWP values derived from observations in the zenith and off-zenith directions. The most important result of the mentioned study is that the LWP retrievals from the ground-based MW measurements definitely demonstrate the existence of the LWP land-sea difference during all seasons and this difference is positive as in case of the satellite measurements (larger LWP values over land and smaller over sea). At the same time, it has been shown that the problem of

the LWP land-sea contrast detection is a complicated one and further research is needed in order to increase the accuracy of the retrieval method and to find explanations for the revealed discrepancies in the magnitude and temporal behaviour of the LWP land-sea contrast obtained from ground-based, satellite and reanalysis data.

## 1.2 Motivation

Recently, the analysis of the LWP land-sea contrast has been done for several water objects of different size located in

Northern Europe: Gulf of Finland, Gulf of Riga, the Neva River bay, lakes Ladoga, Onega, Peipus, Pihkva, Ilmen, and Saimaa (Kostsov et al., 2021). The input data for this analysis were the LWP values of pure liquid-phase clouds derived from the space-borne observations by the SEVIRI instrument in 2011-2017. The study revealed several interesting features in the long-term and short-term variability of the LWP contrast. For example, there are indications of special conditions on the territory of the Gulf of Finland where in June and July large and moderate positive values of the LWP contrast prevail over

negative ones while in August positive and negative values are much smaller (in terms of absolute quantities) and occur with equal frequency (the so-called "August anomaly"). This result can lead to the conclusion about possible common physical





mechanisms that drive the land-sea LWP difference in the Baltic Sea region at small distances from the coastline. Also, the important finding is the positive trend of the land-sea LWP contrast detected within the time period 2011-2017. Another interesting feature is the diurnal cycle of the LWP land-sea contrast which was detected in June and July while there was no evidence for it in August. All these findings require validation and explanation.

Traditionally, ground-based microwave remote measurements of LWP are considered to be a good tool for validation of observations of LWP by satellite instruments and are used for this purpose in practice. So, the main motivation of the present study is to develop a high-accuracy algorithm for the derivation of the LWP land-sea contrast from ground-based MW measurements which can be used both for investigation of the effects related to the LWP contrast and for the validation of the space-borne data. Ground-based zenith and off-zenith microwave measurements performed near the coastline of the Gulf of Finland at the observational station of St.Petersburg State University by the RPG-HATPRO radiometer are a good basis for such research.

### 1.3 Novelty

Novelty of the present study, if compared to previous work by Kostsov et al. (2020), arises from the necessity to consider in detail all aspects of the inverse problem of the LWP land-sea contrast retrieval from ground-based MW measurements in order to reach highest possible accuracy of the results. The previous work used a regression algorithm for the LWP retrievals. Though this algorithm is widely used in practice, the physical inversion algorithm is superior from the point of the retrieval accuracy and data quality control. The problem of the LWP land-sea contrast assessment implies finding the small difference between large highly variable quantities acquired from observations with different geometry. So the high accuracy of the obtained results is the crucial requirement. In the present study we highlight the following main issues which were not addressed earlier:

- measurement geometry analysis accounting for position of clouds;
- application of the multi-parameter physical inversion algorithm to the problem of the LWP contrast retrieval and the specific setup of this algorithm;
- an atmospheric model for the retrieval algorithm and necessary assumptions;
- analysis of possible error sources;
- the problem of bias correction;
- the sensitivity of zenith and off-zenith measurements to atmospheric parameters at different altitudes (implying all parameters which influence the radiative transfer in the MW spectral region, not only the cloud liquid water).

As a result of the study, we not only describe the retrieval procedure which gives accurate results, but also demonstrate new estimations of the LWP land-sea contrast in the region of the Gulf of Finland and make a comparison with the space-borne data.



## 2 Formulation of the inverse problem

The regression algorithm (linear or quadratic) is widely used for processing data from ground-based microwave observations due to simplicity and computational efficiency as its main advantages. Another algorithm for processing these data is called "physical" or "physical-iterative". This algorithm is based on the linearization and inversion of the radiative transfer equation, usually by an optimal estimation method (Rodgers, 2000). Turner et al. (2007) made a detailed analysis of the applicability of both algorithms and of their combination to the problem of derivation of LWP and IWV from two-channel microwave observations. The main superiority of the physical algorithm over the regression algorithm originates from the fact that this algorithm accounts for the spatial distribution of atmospheric parameters along a line-of-sight and gives more options for data quality control. In particular, the information about temperature in cloud layers helps to reduce the LWP retrieval errors. Thorough comparison of the physical and regression algorithms applied to measurements by the RPG-HATPRO 14-channel radiometer was done by Kostsov et al. (2018a). This comparison revealed several important advantages of the physical algorithm. In particular, the identification of cloud-free periods of time using the criterion of minimal observed variations of LWP values can be successfully done using the results obtained by the physical method. However, this criterion fails for the results obtained by the regression method. As a consequence, the data acquired by physical algorithm can be considered self-sufficient for the estimation of bias from cloud-free measurements in contrast to the data acquired by regression algorithm. Also, it has been found that the response of the regression algorithm to artefacts in the input data is considerably larger than the response of the physical algorithm and since there are problems with the detection of cloud-free periods from the data obtained by regression algorithm, one can come to the conclusion that the applying of the physical algorithm is more preferable. However, the combination of both gives an additional possibility for data quality control.

The applicability of the physical method to the problem of the LWP and IWV retrieval by two-channel radiometers implies that the a priori profiles of pressure, temperature and humidity are available from external data sources and the cloud liquid water content (LWC) profile is assigned in a model form. In the process of solving the inverse problem by the physical method, LWC and humidity profiles are modified in one way or another to deliver minimum to the residual between measured and simulated brightness temperatures. For multi-channel radiometers, all mentioned profiles, including temperature and pressure ones, can be derived from microwave observations simultaneously. Additionally, the microwave measurements can be combined with other measurement data and constraints. This approach is described by Loehnert et al. (2008) and called by the authors the IPT (integrated profiling technique). Kostsov (2015ab) used the term "general approach to solution of multi-parameter inverse problems" and considered a priori information and additional constraints as "virtual measurements" following the idea by Rodgers introduced in the pioneering paper (Rodgers, 1976).

Prior to description of the retrieval algorithm, we briefly present the experimental setup. For going into more details of the experiment, a reader is kindly requested to take a look at the study by Kostsov et al. (2020). The RPG-HATPRO



radiometer is located at the observational station of St.Petersburg State University in the suburbs of St.Petersburg (Russian Federation) at 2.5 km from the coastline of the Gulf of Finland. The angular scanning is done every 20 min in the direction of the Gulf of Finland. The measurement geometry is shown in Fig. 1. The off-zenith lines-of-sight pass over two water bodies. The largest one is the Neva river Bay in the Gulf of Finland which is about 15 km wide in the scan plane. The second water body is a small lake 4 km wide which is located at a distance of 22 km from the radiometer. Fig. 1b shows how the

lines-of-sight at different elevation angles cross the areas over water bodies where liquid water clouds can appear. We assume these areas to extend from 0.3 to 5.5 km altitude. Obviously, an optimal elevation angle depends on the cloud base height (CBH) and cloud vertical extent. The elevation angles from 8.4° to 14.4° seem to be of the first priority since they give an opportunity to detect low clouds just in the middle of the water body of interest (the Gulf of Finland). When selecting the angles for scanning, one should keep in mind the opinion expressed by an expert in the open discussion of the

paper by Kostsov et al. (2020) (https://amt.copernicus.org/preprints/amt-2020-52/amt-2020-52-RC2-supplement.pdf, last access 11 Aug 2021):

> …*given the difficulty to interpret the signal below 5 degree, and the fact that it could be related to the interaction between the surface and the atmosphere, it is better to limit the scan to angles > 10 degrees altogether.*

Also, one should keep in mind the difficulty related to finite field-of-view of the instrument which is 3° and can be critical at

170 small elevation angles when viewing vertically and horizontally inhomogeneous clouds and small clouds.

Now we pass to the description of the retrieval algorithm. First of all, we would like to emphasize that measurements at different elevation angles are treated separately since, prior to any further assumptions, we consider all atmospheric parameters as horizontally inhomogeneous. That is why the inverse problem is formulated with respect to the distribution of unknowns along the line-of-sight. In practice, however, an altitude grid is used and, therefore, the obtained profiles should be

treated not as vertical profiles in geometric meaning but as a kind of "effective" profiles. Obviously, the inverse problem in its general formulation through the radiative transfer equation is the classical strongly underdetermined ill-posed problem which requires a system of constraints. Our retrieval algorithm is based on the "general approach to solution of multi-parameter inverse problems" (Kostsov, 2015b) which uses the concept of "virtual measurements" for all kinds of constraints (Rodgers, 1976).

The joint vector of unknowns is:

$$x^{\mathbf{T}} = \left(a_1, a_2, \dots a_{\mathbf{L}}, b_1, b_2, \dots b_{\mathbf{L}}, c_1, c_2, \dots c_{\mathbf{L}}, d_1, d_2, \dots d_{\mathbf{L}}\right) \tag{1}$$

where $a$, $b$, $c$, $d$ are parameters describing the state and (or) composition of the atmosphere, indices "1, 2, …L" denote spatial coordinates (altitude levels), "T" denotes transposition. In our case the sought parameters are temperature, pressure, absolute humidity and cloud liquid water content but we use universal notation in the formulae for simplicity and for consistency with

185 the paper (Kostsov, 2015b). The system of equations to be solved under the general approach is the following:





$$\begin{cases} \boldsymbol{y}_1 - \boldsymbol{y}_{10} = \mathbf{K}_1(\boldsymbol{x} - \boldsymbol{x}_0\,) \\ \boldsymbol{y}_2 - \boldsymbol{y}_{20} = \mathbf{K}_2(\boldsymbol{x} - \boldsymbol{x}_0\,) \\ \quad\cdots\cdots\cdots \\ \boldsymbol{y}_N - \boldsymbol{y}_{N0} = \mathbf{K}_N(\boldsymbol{x} - \boldsymbol{x}_0\,) \end{cases} \tag{2}$$

where N is the total number of actual and virtual measurements, $\boldsymbol{y}_i$ and $\mathbf{K}_i$ are the vectors of the results of actual and virtual measurements and corresponding linear operators, the zero index denotes the reference values for linearization. The microwave measurements and in situ measurements of temperature, pressure and absolute humidity at the radiometer location are actual measurements while a priori information and other constraints are treated as virtual measurements, see Table 1. A priori information includes mean profiles of temperature, relative humidity, pressure, and cloud liquid water content with corresponding covariance matrices. The hydrostatic equilibrium equation couples temperature and pressure and is incorporated as virtual measurement. The link between the absolute humidity and the relative humidity is made using virtual measurement formalism also. The cost function of the least squares method can be written now as:

$$f_{\mathbf{c}} = \sum_{i=1}^{N} \left(\mathbf{y}_i - \mathbf{y}_{i0} - \mathbf{K}_i(\mathbf{x} - \mathbf{x}_0)\right)^{\mathbf{T}} \mathbf{S}_i^{-1} \left(\mathbf{y}_i - \mathbf{y}_{i0} - \mathbf{K}_i(\mathbf{x} - \mathbf{x}_0)\right) \tag{3}$$

where $\mathbf{S}_i$ are the error matrices corresponding to actual and virtual measurements. It should be noted that the presentation of the cost function as the sum of a number of terms corresponding to specific type of measurements is possible since we assume the errors of measurements of different type being not correlated. Minimum of the cost function corresponds to the solution:

$$x = x_0 + \left(\sum_{i=1}^{N} K_i^{\mathbf{T}} S_i^{-1} K_i\right)^{-1} \left(\sum_{i=1}^{N} K_i^{\mathbf{T}} S_i^{-1}(y_i - y_{i0})\right) \tag{4}$$

In fact, we obtain the solution in the iterative process, but the corresponding expression is not presented for simplicity. The expression for the error matrix (the Fisher information matrix) of the solution is the following:

$$F = \left(\sum_{i=1}^{N} K_i^{\mathbf{T}} S_i^{-1} K_i\right)^{-1} \tag{5}$$

Obviously, when we have at our disposal in Table 1 only microwave measurements ($\boldsymbol{y}_1$, $\mathbf{K}_1$, $\mathbf{S}_1$) and a priori statistics for unknowns ($\boldsymbol{y}_3$, $\mathbf{K}_3$, $\mathbf{S}_3$), the formulae (4) and (5) take the well-known forms of the optimal estimation technique:

$$x = x_0 + \left(K_1^{\mathbf{T}} S_1^{-1} K_1 + S_3^{-1}\right)^{-1} K_1^{\mathbf{T}} S_1^{-1}(y_1 - y_{10}) \tag{6}$$

$$F = \left(K_1^{\mathbf{T}} S_1^{-1} K_1 + S_3^{-1}\right)^{-1} \tag{7}$$

We note that in this case $\mathbf{K}_3$ is the unity operator and $\boldsymbol{y}_3 = \boldsymbol{y}_{30}$.

The formalism introduced by Rodgers (2000) gives a possibility to explore an ill-posed inverse problem in detail:



▪ The diagonal elements of the Fisher matrix represent the retrieval errors of sought parameters at different spatial coordinates.

▪ The Fisher matrix can be decomposed into terms which describe the so-called "retrieval noise" and "smoothing error" which are the contributions to the total error made by different actual and virtual measurements.

▪ Spatial resolution of a solution can be obtained with the help of the averaging kernel matrix.

▪ The trace of averaging kernel matrix is equal to the number of independent pieces of information in the solution $x$ delivered by measurements $y$, this quantity is called DOFS - degrees of freedom for signal.

▪ Also, the averaging kernel matrix provides the information on the total spatial region of the reliable retrieval, i.e. the region where the contribution of actual measurements to the solution exceeds the contribution of a priori information.

The problem of the ground-based microwave remote sensing of temperature, water vapour and cloud liquid water content was analysed with the formalism of Rodgers in a large number of studies. In the present paper, we concentrate only on the last item in the list above which is relevant to the assessment of the sensitivity of measurements to atmospheric parameters at different spatial coordinates. The reason for that is the necessity to compare atmospheric parameters derived from measurements which are made with different observation geometries. If the sensitivity of zenith and off-zenith

measurements to a parameter at a given altitude considerably differs, this fact should be taken into account when interpreting the results. Analysis of the sensitivity can be also helpful in developing and estimating retrieval strategies. In the present study, however, we used not the averaging kernel matrix but another quantity with obvious and clear physical meaning for estimating the sensitivity of MW measurements to the variations of atmospheric parameters at various altitudes; this quantity and the obtained results are presented in Section 3.3.

Though the retrieval algorithm is formulated with respect to vertical profiles of parameters, the quantity of interest is the integral characteristic LWP. This quantity is obtained by applying the integration operator to the corresponding part of the combined vector of the solution:

$$LWP = \begin{pmatrix} r_1 & r_2 & \dots & r_L \end{pmatrix} \begin{pmatrix} w_1 \\ w_2 \\ \dots \\ w_L \end{pmatrix} \tag{8}$$

where $r_i$ are the quadrature coefficients for the altitude grid, $w_i$ are the cloud liquid water content values at the altitude grid.

The estimations of the retrieval errors are obtained from the corresponding block (submatrix) of the Fisher matrix $F_{ww}$ as follows:



$$\sigma_{LWP} = \sqrt{ \begin{pmatrix} r_1 & r_2 & \cdots & r_L \end{pmatrix} \mathbf{F}_{ww} \begin{pmatrix} r_1 \\ r_2 \\ \cdots \\ r_L \end{pmatrix} } \qquad (9)$$

The formulae are applied both to zenith and off-zenith observations. Therefore, the LWP values obtained from off-zenith measurements are automatically converted to vertical columns and can be directly compared to the quantities obtained from
zenith observations.

## 3 Retrieval strategy

The main goal of the study is to estimate with high accuracy the difference between the LWP values obtained from zenith and off-zenith observations averaged over relatively long periods of time in order to explore the effect of the "LWP land-sea contrast" in terms of its mean values. It should be emphasized that comparing results of individual instantaneous zenith and
off-zenith measurements is meaningful only for specially and carefully selected test cases since clouds in general are highly variable atmospheric objects which, in addition, move across a line-of-sight of the radiometer due to wind. Taking into account the 20 min interval between angular scans, this motion can be considered as a random process. The issue about the averaging period has been already discussed by Kostsov et al. (2020). It was found that for the described experiment the minimal time period for averaging is 10 days.

However, the optimal retrieval strategy should be worked out for individual instantaneous observations. This strategy includes, first, the optimal selection of elevation angles and, second, the optimal selection of spectral channels for zenith and off-zenith measurements. The latter is necessary for providing equal sensitivity of observations with different geometry to atmospheric parameters. When developing the retrieval strategy, we tried to avoid considerable modifications of the retrieval algorithm for zenith measurements which had been tuned, tested and used for processing the measurement data during years
of operation of the RPG-HATPRO instrument at the observation site of St.Petersburg State University.

### 3.1 Elevation angles, atmospheric model, and scaling factors

The major requirement to the off-zenith geometry concerns the ability to receive a signal from liquid water clouds over the middle of a water object. Let us examine Fig. 2a. It is quite similar to Fig. 1b but the minor and insignificant second water object is not shown. Besides, three smallest elevation angles are removed from consideration in order to avoid possible
influence of the underlying surface. The elevation angles 30° and 19.2° are removed also since corresponding lines-of-sight go high and escape the area of liquid phase clouds already over the water object. Only two lines-of-sight remain, which can be considered as the most suitable for solving the problem (elevation angles 11.4° and 14.4°). In Fig. 1b, the liquid phase



cloud area over the entire water object is shown as the blue rectangle. The base and top of this area are at 0.3 km and 5.5 km correspondingly. These boundaries are in full agreement with the a priori profile of cloud LWC used in the routine retrievals

with the zenith geometry. In Fig. 2a we introduce a new cloud area called "a cloud area of interest" which spans from 1 km to 4 km altitude and is located at a distance of 3 km from both coastlines. The necessity for the consideration of this new area is explained by the necessity to use a self-consistent atmospheric model for interpreting zenith and off-zenith measurements together. It should be noted that definition of the altitude region, where clouds can appear, is an essential part of the physical retrieval algorithm since this algorithm requires a priori profiles and uncertainties of the cloud LWC despite the fact that the

target parameter is the integral characteristic (LWP). It is well-known that microwave observations are almost insensitive to the spatial distribution of cloud LWC but sensitive to the LWP and one should keep in mind this fact. In the retrieval process, the liquid water profile is adjusted by the algorithm to provide proper LWP value, but analysing the resulting LWC profile itself makes no sense.

In order to better understand why the cloud area model used for routine processing of zenith measurements (0.3-

5.5 km altitude) has been changed to the new one (1-4 km altitude), let us consider three situations with different cloud base heights which are schematically shown in Fig. 2a by "Cloud 1", "Cloud 2", and "Cloud 3". According to the new model, clouds can appear in three altitude ranges: $R_{low}$ (0-1 km), $R_{medium}$ (1-4 km), and $R_{high}$ (4-10 km). Let us assume that for a certain long period of observations the total number of measurements is $N$. We can write:

$$N = N_{low} + N_{medium} + N_{high} + N_{clear}$$ (10)

where $N_{low}$, $N_{medium}$, $N_{high}$ are the numbers of cases when clouds appear in the low, medium and high altitude ranges correspondingly, and $N_{clear}$ is the number of simultaneous clear-sky scenes over land and over sea. In Fig. 2a the clouds over the radiometer are not shown but we keep in mind that they appear everywhere. According to our model, the cloud base height (CBH) for clouds over land is the same as over sea and multi-layer cloudiness is not considered. For the mean LWP contrast $D_{mean}$ we can write:

$$D_{mean} = \frac{1}{N} \sum_{i=1}^{N} D_i =$$


$$= \frac{1}{N} \left( \sum_{i=1}^{N_{low}} D_i + \sum_{j=1}^{N_{medium}} D_j + \sum_{k=1}^{N_{high}} D_k + \sum_{m=1}^{N_{clear}} D_m \right)$$ (11)

where $D$ with indices $i$, $j$, $k$, $m$ designates individual measurements of the LWP contrast, i.e. the difference between individual LWPs obtained from zenith (*zen*) and off-zenith (*offz*) instantaneous observations:

$$D = LWP_{zen} - LWP_{offz},$$ (12)

It is evident that there is no LWP contrast under clear-sky conditions: $D_m=0$. Therefore, Eq. 11 takes the form:



$$D_{mean} = \frac{1}{N}\sum_{i=1}^{N_{low}} D_i + \frac{1}{N}\sum_{j=1}^{N_{medium}} D_j + \frac{1}{N}\sum_{k=1}^{N_{high}} D_k =$$

$$= \frac{N_{low}}{N}\frac{1}{N_{low}}\sum_{i=1}^{N_{low}} D_i + \frac{N_{medium}}{N}\frac{1}{N_{medium}}\sum_{j=1}^{N_{medium}} D_j + \frac{N_{high}}{N}\frac{1}{N_{high}}\sum_{k=1}^{N_{high}} D_k = \quad (13)$$

$$= \frac{N_{low}}{N} D_{low} + \frac{N_{medium}}{N} D_{medium} + \frac{N_{high}}{N} D_{high}$$

where $D_{low}$, $D_{medium}$ and $D_{high}$ are the mean values of the LWP contrast calculated separately for three considered cases of cloud location with different CBH. The core idea of our consideration is the following: the values $D_{low}$ and $D_{high}$ obtained from our measurements are equal to zero since clouds in the low and high altitude ranges are intersected by the zenith and off-zenith lines-of-sight over land and just over the coastline where there are no physical reasons for the LWP contrast. Therefore, while Eq. 13 describes the true value of the LWP contrast, the value obtained from measurements is the following:

$$D_{mean}^{measured} = \frac{N_{medium}}{N} D_{medium} \quad (14)$$

The true and measured values relate as:

$$D_{mean}^{true} = D_{mean}^{measured} + \frac{N_{low}}{N} D_{low} + \frac{N_{high}}{N} D_{high} \quad (15)$$

It follows from Eq. 15 that the true value of the LWP land-sea contrast is always larger than the value which we obtain, assuming that the sign of the contrast values is the same for clouds with different CBH. Naturally, the question arises: what useful information can we derive from our measurements then? The answer to this question is the following. First of all, we can obtain the estimation of the LWP contrast $D_{medium}$, which describes only the cloudy scenes with CBH in the range 1-4 km. If we introduce the scaling factor as the ratio of the total number of all scenes (cloudy and clear-sky) to the number of scenes when clouds appear in the altitude range $R_{medium}$:

$$F_1 = \frac{N}{N_{medium}} \quad (16)$$

where index "1" stands for the "first" scaling factor, we can write:

$$D_{medium} = D_{mean}^{measured} F_1 \quad (17)$$

It is obvious that if all scenes are cloudy and all clouds appear in the medium altitude range, the scaling factor is equal to unity. There are no measurements of cloud base and vertical extent at the observational site where the RPG-HATPRO





radiometer is installed. Therefore, we have to use available statistics in order to estimate the ratio of $N$ to $N_{\text{medium}}$ (see Section 4). Second, we can derive the estimate of the initially sought value of the LWP contrast for all scenes. This estimate is very important since it can be compared to the results obtained from space-borne measurements of the LWP land-sea

contrast. However, to achieve this goal, one additional assumption should be made. If we assume that the mean values of the LWP contrast for cloud scenes with different CBH are similar:

$$D_{medium} = D_{low} = D_{high} \tag{18}$$

and if we designate the estimation of true mean value in Eq. 15 as $D_{\text{all}}$, we can rewrite Eq. 15 as follows:

$$D_{all} = D_{mean}^{measured} + \frac{N_{low}}{N} D_{mean}^{measured} F_1 + \frac{N_{high}}{N} D_{mean}^{measured} F_1 =$$
$$= D_{mean}^{measured} \left( 1 + \frac{N_{low}}{N_{medium}} + \frac{N_{high}}{N_{medium}} \right) =$$
$$= D_{mean}^{measured} \frac{N_{medium} + N_{low} + N_{high}}{N_{medium}} =$$
$$= D_{mean}^{measured} \frac{N_{cloudy}}{N_{medium}} = D_{mean}^{measured} F_2 \tag{19}$$

where $N_{\text{cloudy}}$ is the total number of cloudy scenes, and $F_2$ is the second scaling factor (index "2" stands for the "second"):

$$F_2 = \frac{N_{cloudy}}{N_{medium}} \tag{20}$$

This scaling factor, along with $F_1$, can be derived from available statistics (see Section 4).

       Concluding this subsection, for better understanding of the problem of the LWP land-sea contrast detection, we present Fig. 2b where all three described situations of cloud location are shown in more detail. For illustrative purpose, the

true horizontal and vertical scales are not preserved. For case 1, the zenith and off-zenith lines-of-sight (LOS) intersect the clouds over land, and these clouds have the same LWP, so the LWP land-sea contrast is not detected despite the fact that it actually exists (as shown by light blue clouds over the water body and deep blue clouds over land). The same is true for case 3, when zenith and off-zenith LOS intersect clouds over land but over the opposite shores of the water body. And only in case 2 we detect the LWP land-sea contrast since the off-zenith LOS intersect a cloud over sea and the zenith LOS

intersect a cloud over land. So, we have shown that the problem of the LWP land-sea contrast detection from measurements at 2.5 km distance from the coastline can not be solved for very low clouds and for high clouds. If we assumed the existence of the cloud area below 1 km ($R_{\text{low}}$), and higher 4 km ($R_{\text{high}}$) we would definitely put inconsistency in our model since the model should describe off-zenith observations of clouds over water object pretty far from the coastline where the LWP





contrast effect is expected to be maximal. It should be noted that the atmospheric model for zenith observations has been
changed accordingly (the cloud altitude range has been changed to $R_{\text{medium}}$). This modification is a minor one, so we
managed to keep the algorithm for processing zenith observations almost unchanged.

### 3.2 Error sources

Let us analyse qualitatively error sources, except the uncertainty of the scaling factors. The results of the LWP retrievals
from zenith and off-zenith instantaneous observations can be written as:

$$LWP_{zen} = LWP_{zen}^{true} + \varepsilon_{zen}^{ran} + \varepsilon_{zen}^{bias} + \varepsilon_{zen}^{alt}$$

(21)

$$LWP_{offz} = LWP_{offz}^{true} + \varepsilon_{offz}^{ran} + \varepsilon_{offz}^{bias} + \varepsilon_{offz}^{alt} + \varepsilon_{offz}^{FOV}$$

(22)

where *true* stands for a true value, $\varepsilon$ designates an error. Along with the conventional random error (*ran*) and bias (*bias*),
there is an error (*alt*) stipulated by discrepancy between the true position of a cloud and a priori assignment of a cloud area
altitude range. In fact, this error is always present if there is no measurement of CBH and cloud vertical extension or this
measurement exists but it is not accounted for in the retrieval algorithm. In this case the true unknown liquid water profile is
mapped into the a priori cloud altitude range and this mapping causes the error component $\varepsilon^{alt}$. Off-zenith observations
include, in addition, the error due to finite field-of-view (FOV) of the instrument. This error is expressed as a specific term in
Eq. 22 and requires consideration for off-zenith geometry because the distance from the instrument to a cloud for off-zenith
observations can be much larger than for zenith observations (by a factor of 4-5 for selected elevation angles). As a result,
for small elevation angles, the signal which comes to the instrument is formed in a quite large area inside a cloud or both
inside and outside a cloud in case of small clouds. The difficulties relevant to this issue have been earlier revealed by an
expert in the open discussion of the paper by Kostsov et al. (2020) (https://amt.copernicus.org/preprints/amt-2020-52/amt-
2020-52-RC2-supplement.pdf):

>     … *the instrument field of view (3 degrees) makes it difficult to interpret the off-zenith measurements if the cloud*
>     *boundaries are not known. With a 3-degree FOV the radiometer will be sampling a horizontal area of ~ 1km at 20 km*
>     *distance when looking up.*

The LWP contrast *D* estimated as an average of multiple observations can be expressed as:

$$D_{mean}^{measured} = \frac{1}{N}\sum_N \left(LWP_{zen} - LWP_{offz}\right) =$$

$$= D_{mean}^{true} + \frac{1}{N}\sum_N \left(\varepsilon_{zen}^{ran} - \varepsilon_{offz}^{ran}\right) + \varepsilon_{zen}^{bias} - \varepsilon_{offz}^{bias} + \frac{1}{N}\sum_N \left(\varepsilon_{zen}^{alt} - \varepsilon_{offz}^{alt}\right) - \frac{1}{N}\sum_N \left(\varepsilon_{offz}^{FOV}\right)$$

(23)



In Eq. 23, the term containing the random error component is negligibly small due to averaging. The term containing the
error component due to vertical misplacement of a cloud is also negligibly small because of the following reason: this
component for instantaneous and simultaneous zenith and off-zenith observations is expected to be of the same magnitude,
so their difference is expected to be close to zero. Extensive analysis of the error component caused by finite FOV has been
done in the study by Kostsov et al. (2020) on the basis of numerical simulations: the difference between the brightness
temperature calculated neglecting FOV and the brightness temperature calculated accounting for FOV was used as a measure
which characterises in the best way the FOV influence on the results of the interpretation of the off-zenith measurements.
The numerical simulations have shown that the mean value of this difference is less than 0.4 K for the two elevation angles
selected in the present study and this value can be considered as negligible. The obtained values of the standard deviation of
this difference have been used for the estimation of a minimal number of individual measurements which should be sampled
in order to suppress considerably the influence of FOV. Averaging over a 10 day time period has been found to be sufficient
for suppressing the random error due to FOV. So, now we can write the final formulae which should be used for obtaining
the estimates of two quantities which characterise the mean value of the LWP land-sea contrast and which have been
described above:

$$D_{medium} = \left( D_{mean}^{measured} - \varepsilon_{zen}^{bias} + \varepsilon_{offz}^{bias} \right) F_1 \tag{24}$$

$$D_{all} = \left( D_{mean}^{measured} - \varepsilon_{zen}^{bias} + \varepsilon_{offz}^{bias} \right) F_2 \tag{25}$$

The values of bias for zenith and off-zenith observations can be obtained from measurements under clear sky conditions
using, for example, the approach by Cossu et al. (2015) who fitted the distribution of obtained LWP values under clear sky
conditions with a single-term Gaussian model.  The standard deviation gives in this case the random error and the mean
value gives the bias of the LWP retrievals. The scaling factors $F_1$ and $F_2$ can be assessed from statistics acquired at the
neighbouring observational meteorological stations, if available, or from reanalysis data. It looks reasonable to consider and
apply monthly mean or seasonally mean values of the scaling factors in Eqs. 24 and 25.

**3.3 Retrieval setup and sensitivity functions**

We use the term 'retrieval setup' to designate a set of data which have to be prepared for proper functioning of the retrieval
algorithm for all selected elevation angles. This set can include a priori information about sought parameters, characteristics
of an instrument or of several instruments (in case of combined observations), measurement error estimations, an upper limit
for iterations, a convergence criterion for solution, quality control criteria, etc.  It should be emphasised that the core idea of
the current study is to analyse not the quantity LWP itself but the difference between the corresponding values obtained from
zenith and off-zenith observations. Also, it should be noted that the integral parameter LWP is derived from the retrieved
vertical profile of LWC. Therefore, for accurate assessment of the difference it is important to ensure that the sensitivity of
microwave measurements to cloud liquid water content at a given altitude is approximately the same for all selected





geometries. We have this requirement also for absolute humidity and temperature. If the sensitivity is practically the same, then the systematic error component which is always present in the retrieval will be cancelled when subtraction will be executed.

For assessment of the sensitivity of microwave measurements to atmospheric parameters, we used the information gain function which has very simple physical meaning: the relative decrease of the a priori uncertainty of a sought parameter

in the process of solving the inverse problem. This function is defined as follows:

$$G(z) = \frac{\left(\sigma_{apr}(z) - \sigma_{apost}(z)\right)}{\sigma_{apr}(z)} 100\% \tag{26}$$

where $z$ is altitude, $\sigma_{apr}$ is the a priori uncertainty of a given parameter in terms of a priori standard deviation, $\sigma_{apost}$ is the a posteriori uncertainty, i.e. the retrieval error. Obviously, for altitude regions with very large retrieval errors, this information gain function will be near zero, while the regions with accurate retrievals will be characterized by high values of

$G$. When we have a priori information of different kind and follow the idea of actual and virtual measurements, it is useful to consider three types of the gain function:

$$G_1(z) = \frac{\left(\sigma_{apr}(z) - e_1(z)\right)}{\sigma_{apr}(z)} 100\%$$

$$G_2(z) = \frac{\left(\sigma_{apr}(z) - e_2(z)\right)}{\sigma_{apr}(z)} 100\% \tag{27}$$

$$G_3(z) = \frac{\left(e_2(z) - e_1(z)\right)}{e_2(z)} 100\%$$

where $e_1$ is the retrieval error in case when a complete set of actual and virtual measurements is used, $e_2$ is the retrieval error when all actual and virtual measurements are used except the microwave measurements. Physical meaning of these functions

is straightforward. $G_1$ characterises the retrieval algorithm as a whole, $G_2$ gives the impression how much information is delivered by in situ measurements at the radiometer position and by constraints, and $G_3$ describes how much information is delivered by microwave measurements with respect to the information provided by in situ measurements and constraints. It is self-evident that $e_2$ is always larger than $e_1$ or equal to it.

The results of calculations of three gain functions for each atmospheric parameter (temperature, absolute humidity and

cloud liquid water content) are presented in Fig. 3 for two elevation angles: 90° and 11.4°. First of all, we note that functions $G_2$ for temperature and humidity show a maximum at the ground level and illustrate the result of statistical extrapolation of the ground-level in situ measurements of temperature and humidity. The $G_2$ function for cloud LWC is equal to zero reflecting the fact that we obtain the information on LWC only from microwave measurements. Therefore, $G_3=G_1$ for LWC. The considerable difference between $G_2$ and $G_1$ and the large values of $G_3$ are an indication of the considerable contribution

of the microwave measurements to the solution with respect to other measurements and constraints. For comparing the gain





functions for different geometries, we shall use $G_3$. There are two important conclusions which can be derived from such a comparison:

1) The $G_3$ functions for absolute humidity and LWC have similar values for both geometries. The similar sensitivity of the microwave measurements to these parameters for zenith and off-zenith geometry is explained by the transparency of the so-called 'humidity spectral channels' of the radiometer (7 channels in the range 21-32 GHz). When the elevation angle gets smaller, the optical path increases, however these channels still remain transparent enough for sensing upper tropospheric layers.

2) The $G_3$ functions for temperature noticeably differ for zenith and off-zenith geometry due to the fact that the 'temperature spectral channels' (7 channels in the range 50-60 GHz) are opaque, especially high frequency ones. As a result, for off-zenith geometry we have the increase of sensitivity at lower altitudes and the decrease of sensitivity at upper altitudes.

As one might think, the retrieval setup needs no modification since we have practically the similar sensitivity to absolute humidity and cloud liquid water for the zenith and off-zenith geometry. However, this conclusion will be altered if we recall that we solve the multi-parameter inverse problem and the retrieval results for temperature, humidity and LWC have cross-links because of constraints and because of physics of radiative transfer (absorption of water vapour and cloud droplets is temperature dependent). Measurements in highly opaque temperature channels 'probe' the air volumes near the radiometer at low altitudes. This is the region over the roof of the building where the radiometer is installed and where noticeable temperature horizontal inhomogeneities can occur due to heating of the roof by solar irradiance in winter and summer and by the internal heating of the building in winter. In this case the in-situ temperature and humidity measurements at the radiometer position will not be in agreement with the neighbouring air. Such a situation can cause large retrieval errors for temperature and, in turn, for humidity and LWC. Therefore, we come to the conclusion that we have to try to modify the original retrieval setup in order to obtain the gain functions for all parameters and for zenith and off-zenith geometry as similar as possible.

For changing the gain functions $G_3$, we used a simple approach with clear physical meaning. The 'weight' of measurements in a given spectral channel depends on the measurement error. By assigning a large or extremely large measurement error value, we can partly or completely switch off a particular channel. We tried several combinations and finally found the optimal combination which is presented in Table 2. The original setup with measurement errors equal to 0.1 K and 0.2 K in the humidity and temperature channels respectively has been modified in the following way:

1) In order to decrease the sensitivity to absolute humidity for off-zenith observations, the measurement errors in channels 1-6 were assigned the value 0.3 K.

2) In order to decrease the sensitivity to LWC for off-zenith observations, the measurement error in channel 7 was assigned the value 0.35 K.





3) In order to decrease the sensitivity to temperature for off-zenith observations in the lower layers, the channels 12-14 were switched off and the measurement error in channel 11 was assigned the value 0.5 K.


4) In order to decrease the sensitivity to temperature for zenith observations in the upper layers, the channel 10 was switched off and the measurement error in channel 9 was assigned the value 0.5 K.

One comment to the list above is needed: it is evident, that for a given instrument there is no way to increase the sensitivity (to make the sensitivity functions larger).

Fig. 4 illustrates the results of the modification of the original retrieval setup. The upper row demonstrates noticeable

difference of the sensitivity functions for zenith and off-zenith geometry if the original setup is used. The lower row of panels displays an excellent agreement of the sensitivity functions in case we use the modified retrieval setup. It should be emphasised that we managed to obtain this agreement for all target parameters: temperature, absolute humidity and cloud LWC.

## 4 Assessment of the scaling factors

To estimate the scaling factors $F_1$ and $F_2$ which have been introduced in section 3.1, we analysed archived weather observation data, namely the values of the cloud base height (CBH). We used the archives of observations performed at three meteorological stations. One station is located in St.Petersburg (https://rp5.ru/Weather_archive_in_Saint_Petersburg, last access 28 November 2021) at 27 km distance from the RPG-HATPRO radiometer. Two other stations are located not far from the position of the RPG-HATPRO instrument: "Lomonosov" (https://rp5.ru/Weather_archive_in_Lomonosov, last

access 28 November 2021) and "Kronstadt" (https://rp5.ru/Weather_archive_in_Kronstadt, last access 28 November 2021). The "Lomonosov" station is located on the very shore of the Neva Bay, 6 km northwest of RPG-HATPRO. The "Kronstadt" station is located on Kotlin Island, approximately in the middle of the Neva Bay and 14 km north of RPG-HATPRO. The available data records contain the values of CBH of the lowest clouds (in meters), routinely estimated by an observing person every 3 hours. These estimates are given in the records as belonging to the following classes (or ranges): 100-200 m,

200-300 m, 300-600 m, 600-1000 m, 1000-1500 m, 1500-2000 m, 2000-2500 m, 2500 m and higher. For simplicity, we considered the lower boundary of each class as the observed CBH of the lowest clouds, i.e. 100, 200, 300, 600, 1000, 1500, 2000 and 2500 meters. To obtain the required statistical information about the average vertical distribution of the cloud base, we analysed data series of continuous observations at all three stations in the period from 2011 to 2017. We selected only the daytime data for the purpose of compliance with the observations by the SEVIRI satellite instrument (the comparison of the

LWP contrast obtained from the ground-based and space-borne measurements will be discussed below in Section 6). Analysis of the long-term records has shown that the complete data set is provided by the St.Petersburg station while the data records from Kronstadt and Lomonosov have gaps in which several parameters are missing. These data gaps make correct identification of clear-sky scenes impossible, so the data from stations in Lomonosov and Kronstadt were used only for





calculation of the factor $F_2$. The frequency of occurrence of each class of CBH for the St.Petersburg station is shown in Fig.

5. According to this statistics, relatively low clouds prevail in the daytime in St. Petersburg. The clouds with the CBH less

than 1000 meters taken together occur in 62% of cases. Higher clouds with CBH equal to 1000 m or higher occure in 28% of

the cases while clear-sky scenes are rather rare with a frequence of occurrence of 10%.

To estimate the value of $N_{\text{medium}}$ in Eqs. 16 and 20, we define it as:

$$N_{\text{medium}} = N_{1000} + N_{1500} + N_{2000} + N_{2500} \quad , \tag{28}$$

where $N_{1000}$, $N_{1500}$, $N_{2000}$, and $N_{2500}$ are the numbers of cases when cloud base is observed at altitudes of 1000, 1500, 2000 and

2500 meters, respectively. Since $N_{2500}$ includes all cases with a cloud base height above 2500 meters (including those above

the level of 4000 meters), the $N_{medium}$ value obtained in this way will be overestimated. Thus, we can get the lower estimates

of $F_1$ and $F_2$ from the expressions below:

$$F_1 = \frac{N}{N_{1000} + N_{1500} + N_{2000} + N_{2500}} \tag{29}$$


$$F_2 = \frac{N_{cloudy}}{N_{1000} + N_{1500} + N_{2000} + N_{2500}} \tag{30}$$

where $N$ is the total number of all cases (cloudy and clear-sky), $N_{\text{cloudy}}$ is the number of cloudy cases. Resulting monthly

values of $F_1$ and $F_2$ for daytime are shown in Fig. 6. Comparison of $F_1$ and $F_2$ obtained from the St.Petersburg station records

is given in Fig. 6a and demonstrates very similar values for all months. Minimum monthly values are observed in spring and

summer, maximal values – in autumn and winter. Comparison of scale factor $F_2$ derived from the records of all three

meteorological stations is shown in Fig. 6b. One can see that the main feature of intra-annual variability is the same for all

stations: minimal values in spring and maximal in late autumn. Apparently, this behavior is due to the predominance of the

lowest clouds in late autumn, and of the highest clouds – in spring. But there are also noticeable differences. First, while $F_2$

for the St.Petersburg station is nearly constant from March to September, $F_2$ for the Kronstadt and Lomonosov stations

increases during this period. From December to April, the values of $F_2$ for all three stations are very similar. From May to

November, there is noticeable difference between the values obtained at St.Peterburg station and two other stations. For

Kronstadt and Lomonosov, the scaling factor is approximately 1.5-2.5 times higher than for St.Petersburg. This indicates the

considerable uncertainty which can be present in the values of scaling factor.

## 5 Assessment of the LWP retrieval bias

The LWP retrieval bias originates from instrumental drifts, uncertainties in the retrieval assumptions and variable water

vapor influence on absorption at 31 GHz (Meijgaard and Crewell, 2005; Matzler and Morland, 2009). Usually, the

assessment of the LWP retrieval bias is obtained from clear sky observations. The best conditions for such an assessment are



identified by looking at the standard deviation of retrieved LWP for a certain period of time. The smallest standard deviation values indicate a period where cloud liquid water is totally absent in the atmosphere. The problem of identification and analysis of cloud-free periods has been analysed in detail by Matzler and Morland (2009). As it has already been mentioned

in section 3.1, for the assessment of the bias, the distribution of LWP values in clear-sky conditions is usually fitted with a single-term Gaussian model where the mean value gives the bias (Cossu et al., 2015).

In the present study we used the same approach, but with a slight modification. Since the angular scans are performed by the radiometer rarely (every 20 min), it is not possible to identify the clear sky periods using the LWP standard deviation criterion. Therefore, we selected the data with the minimal LWP values detected by zenith observations rather than with the

minimal standard deviation. The critical point for the bias assessment is that simultaneously, for corresponding off-zenith observations, the clear sky conditions should also have been present over the water body. Obviously, there is no way to check if it was so in reality. However, it is reasonable to assume that if the criterion based on a certain upper limit for LWP is applied to the results of the off-zenith observations, then such cases could be attributed to clear sky conditions. So, the algorithm was the following:

1)  First, an atmospheric scan was selected for further analysis if the LWP value derived from zenith observation was not larger than $0.005\,\mathrm{kg\,m^{-2}}$. If not, the atmospheric state was marked as cloudy and not applicable for bias assessment.

2)  Second, for scans selected at the first stage, the LWP values derived from off-zenith observations at the elevation angles 14.4° and 11.4° were checked. If both of them did not exceed the LWP value derived from zenith

observation by more than $\delta = 0.020\,\mathrm{kg\,m^{-2}}$ than this angular scan was kept for bias assessment. The value $0.020\,\mathrm{kg\,m^{-2}}$ was obtained from series of tests which showed that further increasing of $\delta$ does not change the result of bias assessment.

3)  The mean LWP values for zenith and off-zenith measurements of selected angular scans were taken as the corresponding bias values.

The described algorithm of the bias assessment was applied to data sets acquired within monthly periods in order to provide consistency with the time periods of the assessment of the LWP land-sea contrast. The obtained bias values are presented in Fig. 7 in the form of a bar chart. In many cases the LWP retrieval bias for off-zenith measurements is noticeably larger than for zenith measurements and constitute from $0.008\,\mathrm{kg\,m^{-2}}$ to $0.019\,\mathrm{kg\,m^{-2}}$. This result is expected since for off-zenith geometry the detected signal is large due to longer optical path and hence the water vapor influence on the absorption

is great. However, there are many situations (2013, 2014, 2016, June 2020, and July 2021) when bias values for zenith and off-zenith geometry are almost the same. The analysis of possible reasons for temporal variations of bias is beyond the scope of the present study. Nevertheless, in order to validate the obtained bias estimations we applied the algorithm of bias assessment to several short time periods when the clear sky conditions were confirmed by visual control. These periods were the days when ground-based spectroscopic Fourier-transform infrared (FTIR) measurements of direct solar radiation have



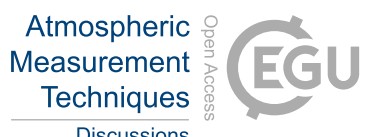

been carried out at the St.Petersburg University measurement site of NDACC (Network for the Detection of Atmospheric Composition Change) (http://www.ndaccdemo.org/stations/st-petersburg-russian-federation, last access 1 October 2021). Two examples of LWP measurements at these days are shown in Fig. 8. One can see that even for angular scan measurements with long sampling intervals the standard deviation of LWP is very small and constitutes about 0.001-0.002 kg m$^{-2}$. Along with visual control of the sky, this is the direct indication of the clear sky conditions according to the "classical" criterion used in microwave radiometry. One can see from Fig. 8 that the bias values for zenith and off-zenith LWP measurements are in excellent agreement with the mean monthly values shown in Fig. 7 (August 2014 and July 2017).

## 6 Results of the estimation of the LWP land-sea contrast

The averaging of the individual measurements of the LWP land-sea difference was done over monthly periods. For analysis, the summer months were selected within the observation period 2013-2021. Such selection was motivated by the fact that previously the LWP land-sea difference derived from the space-based observations by the SEVIRI instrument was thoroughly analysed exactly for summer months (Kostsov et al., 2021). The algorithm of computations and quality control was the following:

1) The LWP retrieval bias for zenith and off-zenith observations was assessed on a monthly basis as described in section 5.

2) The zenith and off-zenith observations (elevation angles 90°, 14.4°, and 11.4°) were processed by the physical inversion algorithm with the retrieval setup described in section 3.3. For the sake of comparison with the space-borne data provided by the SEVIRI instrument, only the microwave observations corresponding to solar zenith angle smaller than 72° were collected for analysis. (It should be noted that SEVIRI acquires data only under sun illumination conditions).

3) Quality control procedures were applied to the results of the LWP retrieval. The results without successful convergence of the iterative retrieval process (no convergence within 12 iterations) were filtered out. The second quality control check was related to the spectral residual, i.e. the discrepancy between measured and calculated brightness temperatures. The measurements with the RMS discrepancy for the humidity channels larger than 0.5 K were removed from further analysis. For temperature channels, the limit 1.0 K was taken for the RMS discrepancy since the residual for the temperature channels is always noticeably larger than for the humidity channels. The quality control analysis is an essential part of the retrieval process and is aimed to avoid the influence of rain events over land or sea, or other possible interfering factors, on the retrievals.

4) Bias correction was applied to the LWP retrieval results.

5) The LWP land-sea difference for an individual measurement was calculated using LWP retrievals at three elevation angles of a single angular scan: 90° (zenith geometry), 14.4° and 11.4° (off-zenith geometry). As a





result, two values of LWP difference were obtained: $D_1$ and $D_2$, which correspond to off-zenith observations at 14.4° and 11.4° respectively, see Eq. 12. These individual differences were averaged over a monthly period.

6) The averaged values of the LWP land-sea difference were multiplied by the scaling factor (see sections 3.1 and 4) in order to obtain the estimation of the true value of the LWP land-sea contrast.

The results of the retrieval of the LWP land-sea difference for summer months within the period 2013-2021 are shown in Fig. 9 in the form of a bar chart. We emphasise, that no scaling factors are applied to the values which are presented. The measurements during 27 month total were processed. For August 2015 the retrieval results did not pass the quality control due to occasional miscalibration of the instrument, and in August 2016 there were problems with power supply which resulted in a loss of measurements. So, finally we had 25 months of high quality data at our disposal. Only for one out of 580 twenty five months (June 2016), the LWP land-sea difference $D_2$ is equal to zero. For all other 24 months out of 25 months, both values $D_1$ and $D_2$ of the LWP land-sea difference are positive. This result is in a good qualitative agreement with the space-borne data which demonstrate positive differences for a warm season when cold water cools the near surface air and makes the atmosphere stable over a water body. This stability prevents convection and formation of clouds. The values differ from year to year and from month to month. There is no any definite intra-seasonal dependence of the LWP difference. 585 The magnitude of the difference varies considerably: from 0.001-0.002 kg m$^{-2}$ to 0.011-0.012 kg m$^{-2}$. The low values of the measured LWP land-sea difference are not a surprise: it has been shown above that because of specific geometry and cloud positions the measured values are always smaller than the true values. We also would like to note that:

- small measured values of the LWP difference are the evidence of the complexity of the problem which we try to solve;

- small measured values of the LWP difference indicate that the results must be very sensitive to bias correction and data quality control procedures.

The latter conclusion was confirmed by tests in a simple way by switching off the quality control.

The true LWP land-sea contrast was assessed by applying the scaling factor to measured quantities. The results are presented in Fig. 10 in the form of a bar chart similar to Fig. 9. The following values of the scaling factor $F_2$ were applied: 595 5.3, 5.8, and 7.4 for June, July, and August respectively. These values are the average scaling factors derived from the data obtained at the two meteorological stations Lomonosov and Kronstadt (see section 4 and Fig. 6b) which are in the closest vicinity to the radiometer position. Since scaling factors are different for June, July and August, Fig. 10 does not represent a simple change of the vertical scale of axes in Fig. 9. After scaling, the values for August became larger with respect to the values for June and July. Also, one can see that the estimates of the true LWP land-sea contrast, obtained by applying scaling 600 factors to the measured quantities, are quite large and can reach very high values up to 0.06-0.07 kg m$^{-2}$. This result is in a good quantitative agreement with the space-borne data obtained for different locations in Northern Europe from the SEVIRI instrument (Kostsov et al., 2021). We again pay attention to the fact that no definite intra-seasonal features of the LWP contrast are revealed. For every summer month, one can find the low LWP contrast values and the high ones as well. The



examples of such pairs of low-high values are 2015 and 2016 for June, 2017 and 2013 for July, and 2017 and 2014 for
August.

In Fig. 11 we demonstrate a comparison of the results of the assessment of the LWP land-sea contrast based on
ground-based microwave measurements with the results of the satellite observations by the SEVIRI instrument. The satellite
data were taken from the study by Kostsov et al. (2021). In the mentioned study, the time period 2011-2017 was analysed
which partly overlaps with the time period considered in the present work (2013-2021). So, we compare the mean monthly
values of the LWP contrast averaged over these overlapping 8-year and 9-year periods. The values $D_1$ and $D_2$ obtained from
ground-based measurements have been averaged jointly and the same scaling factor $F_2$ was applied: 5.3, 5.8 and 7.4 for
June, July and August respectively.

First, the so-called "August anomaly" should be mentioned which was revealed by the satellite observations and
shows up as the practically total absence of the LWP contrast in August if compared to June and July. It should be
emphasised that this "August anomaly" concerns the Gulf of Finland only; this effect is absent for neighboring large and
small lakes (Kostsov et al., 2021). One can see that the ground-based measurements produce no evidence of this anomaly:
the LWP land-sea contrast values for all summer months are almost the same. Moreover, the highest values are detected in
August. For June and July, the results obtained from satellite observations are higher than the results obtained from ground-
based measurements by 0.008-0.009 kg m$^{-2}$. To our opinion, the agreement between the space-borne and ground-based
results for these two months can be estimated as very good. The reason for this conclusion is the great influence of the
scaling factor $F_2$ (described in section 4) on the results of the assessment of the LWP land-sea contrast. First of all, the
estimates of $F_2$ obtained in section 4 were the lower ones. In reality, $F_2$ can be larger and in this case the discrepancy
between the ground-based and the satellite data for June and July will be smaller. Second, we applied the mean values of the
scaling factor to the ground-based data for all months of observations. The variability of true values of $F_2$, which is not taken
into account, is a source of additional error which is present in the ground-based data.

Finally, we would like to make some speculations on the problem of the error budget assessment for the results
derived from the ground-based observations. Two sources of errors have been already discussed above: the LWP retrieval
bias and the uncertainty of the scaling factor $F_2$. To our opinion, there is a way to estimate the overall contribution of all
other remaining error sources of unknown origin. We can treat the difference between the $D_1$ and $D_2$ monthly values
(obtained from off-zenith geometry with different elevation angles 14.4° and 11.4°) as an estimation of the errors produced
by different sources except the bias and scaling factor uncertainty. The corresponding formula will be:

$$\sigma_r = \sqrt{\frac{1}{N}\sum_N (D_1 - D_2)^2}$$
(31)





where $\sigma_r$ is the error estimate for remaining error sources of unidentified origin, $N$ is the number of months of measurements. We applied formula (31) to our data and obtained the following values: $\sigma_r$=0.006 kg m$^{-2}$ and $\sigma_r$=0.001 kg m$^{-2}$ for calculations

with and without scaling factor respectively. Obviously, these values are the "first guess" estimations, but they can be a good reference point for analysis of the accuracy of obtained results and first of all they show that the dominant error source is the uncertainty of scaling factor.

## 7 Summary and conclusion

In this study we investigated the problem of detection of the cloud liquid water path (LWP) horizontal inhomogeneities by

means of ground-based microwave observations and analysed several theoretical and practical aspects of this challenging task. The main difficulties are caused by the large variety of cloud size and cloud altitude in combination with specific observation geometries. The study is based on the data acquired by the microwave radiometer RPG-HATPRO which is located in the suburbs of St.Petersburg, Russia, at the observational site of St.Petersburg State University, at 2.5 km distance from the coastline of the Neva Bay (the Gulf of Finland). The radiometer is operating in the angular scanning mode in a

vertical plane and the scanning is being done towards the water body. The goal is to measure the LWP land-sea contrast (the difference between values of LWP over land and over the water body) which was previously observed in this region by satellite instruments. So, the core idea of the study is to analyse not the quantity LWP itself but the difference between the corresponding values obtained from zenith and off-zenith observations. Therefore, the requirements to the accuracy of measurements and to the quality control procedures are strong.

The physical inversion algorithm has been selected as a main tool for solving the inverse problem of the LWP retrieval. The inverse problem has been formulated as a multi-parameter problem with simultaneous retrieval of temperature, humidity and cloud liquid water content (LWC) profiles. The LWP values are obtained by integration of the cloud LWC profiles. The LWC profiles themselves are not analysed because of poor spatial resolution of the ground-based passive microwave observations. The retrieval setup has been specially organised in a way to provide equal sensitivity of zenith and

off-zenith measurements to the vertical profiles of atmospheric parameters. To achieve this goal, a simple approach with clear physical meaning has been used: the 'weight' of measurements in a given spectral channel depends on measurement error, so, by assigning large or extremely large measurement error values, we can partly or completely switch off a particular channel. The optimal combination of 'weighted' channels has been found for zenith and off-zenith geometries. This combination provided equal sensitivity of measurements to profiles of all retrieved parameters: temperature, humidity, and

LWC. As a result, possible systematic errors were minimised.

    The influence of the cloud base height (CBH) on the possibility to detect the LWP land-sea contrast has been analysed in detail. It has been shown that, for current position of the radiometer and for current observation geometries, the values of the LWP land-sea contrast obtained from microwave measurements are strongly underestimated. To correct the values, a





scaling factor should be used. This scaling factor has been assessed from cloud statistics. The assessment revealed seasonal
dependence of the scaling factor with maximum in autumn-winter and minimum in spring-summer. For summer months the
scaling factor is in the range from 4.5 to 7.5.

The LWP land-sea contrast was estimated for summer months within the measurement period 2013-2021. The
averaging of the individual measurements of the LWP land-sea difference was done over monthly periods. The zenith and
off-zenith observations (elevation angles 90°, 14.4°, and 11.4°) were processed by the physical inversion algorithm. The
LWP retrieval bias for zenith and off-zenith observations was assessed on a monthly basis. Quality control procedures were
applied to the results of the LWP retrieval in order to filter out spurious data. We had 25 months of high quality data at our
disposal. Only for one out of twenty five months, the LWP land-sea difference is equal to zero. For all other cases (24 out of
25 months), the LWP land-sea contrast is positive and sometimes can be rather high approaching the value of
0.06-0.07 kg m$^{-2}$. This result is in good general qualitative and quantitative agreement with the space-borne data which
demonstrate large positive difference for a warm season when cold water cools the near surface air and makes the
atmosphere stable over a water body.

Comparison of the estimations of the LWP land-sea contrast obtained by applying the scaling factor to the ground-
based microwave measurements by the RPG-HATPRO radiometer with the results of the satellite observations from the
SEVIRI instrument has been made for summer months. The mean monthly values of the LWP difference averaged over the
8-year (SEVIRI) and 9-year (HATPRO) periods were analysed. For June and July, the results obtained from satellite
observations are 0.041-0.043 kg m$^{-2}$ and the results obtained by ground-based measurements are 0.032-0.035 kg m$^{-2}$, so the
satellite data are higher by about 0.008-0.009 kg m$^{-2}$. For August, the space-borne data demonstrate very small land-sea
contrast, but the ground-based data show a contrast which is even higher than the one detected in June and July. Three main
conclusions of the study can be formulated:

1) The ground-based microwave observations in the vicinity of a coastline can be a valuable tool for validation of
the space-borne measurements of the LWP land-sea contrast if three principal requirements are met: (a) the multi-
parameter physical inversion method is used for retrieving LWP; (b) rigorous bias correction and quality control
procedures are applied to the retrieval results; (c) the information on the cloud base height is available.

   2) The estimations of the LWP land-sea contrast obtained from the ground-based microwave measurements at the
observational site of St.Petersburg University are in very good agreement with the values of the LWP land-sea
contrast obtained from the multi-year space-borne measurements by the SEVIRI instrument in the region of the
Neva Bay (the Gulf of Finland) in June and July.

   3) For August, the so-called "August anomaly" detected by space-borne observations in the Gulf of Finland region
is not confirmed by the ground-based measurements.





### Data availability

The LWP data derived from the RPG-HATPRO observations at the measurement site of Saint Petersburg State University are available upon request (please write to Vladimir Kostsov at v.kostsov@spbu.ru).

### Author contributions

VSK conceived the study, made the cloud liquid water path retrievals from ground-based microwave observations and prepared the draft of the manuscript. DVI was responsible for cloud statistics. VSK, DVI and AK together interpreted and analysed the results, reviewed and edited the manuscript.

### Competing interests

The authors declare that they have no conflict of interest.

### Acknowledgements

The operation of the RPG-HATPRO instrument was provided by the Research Centre GEOMODEL of St. Petersburg State University (http://geomodel.spbu.ru/).

### Funding

This research has been supported by the Russian Foundation for Basic Research through the project No. 19-05-00372.

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





**Table 1.** Application of the actual and virtual measurement concept to the considered inverse problem.

| Type of measurement | Index j | Description | Zenith / Off-zenith geometry |
|---|---|---|---|
| Actual | 1 | Microwave measurements by the RPG-HATPRO radiometer (brightness temperatures) | Measurements differ |
|  | 2 | In situ measurements of temperature, pressure and relative humidity at the radiometer | Measurements are the same |
| Virtual | 3 | A priori profiles of temperature, pressure, relative humidity and cloud liquid water and corresponding covariance matrices | Measurements are the same |
|  | 4 | Hydrostatic equilibrium equation which couples pressure and temperature |  |
|  | 5 | Equation which couples absolute and relative humidity profiles[1] |  |

[1] The equation is applied since absolute humidity is taken as one of unknowns, not relative humidity.

785

**Table 2.** Retrieval setup: spectral channels and measurement errors.

| Humidity channels | | | | | | | |
|---|---|---|---|---|---|---|---|
| Channel No | 1 | 2 | 3 | 4 | 5 | 6 | 7 |
| Frequency, GHz | 22.24 | 23.04 | 23.84 | 25.44 | 26.24 | 27.84 | 31.40 |
| Error, K / zenith | 0.1 | 0.1 | 0.1 | 0.1 | 0.1 | 0.1 | 0.1 |
| off-zenith | 0.3 | 0.3 | 0.3 | 0.3 | 0.3 | 0.3 | 0.35 |
| Temperature channels | | | | | | | |
| Channel No | 8 | 9 | 10 | 11 | 12 | 13 | 14 |
| Frequency, GHz | 51.26 | 52.28 | 53.86 | 54.94 | 56.66 | 57.30 | 58.00 |
| Error, K / zenith | 0.2 | 0.5 | Switched off | 0.2 | 0.2 | 0.2 | 0.2 |
| off-zenith | 0.2 | 0.2 | 0.2 | 0.5 | Switched off | Switched off | Switched off |



(a)

(b)

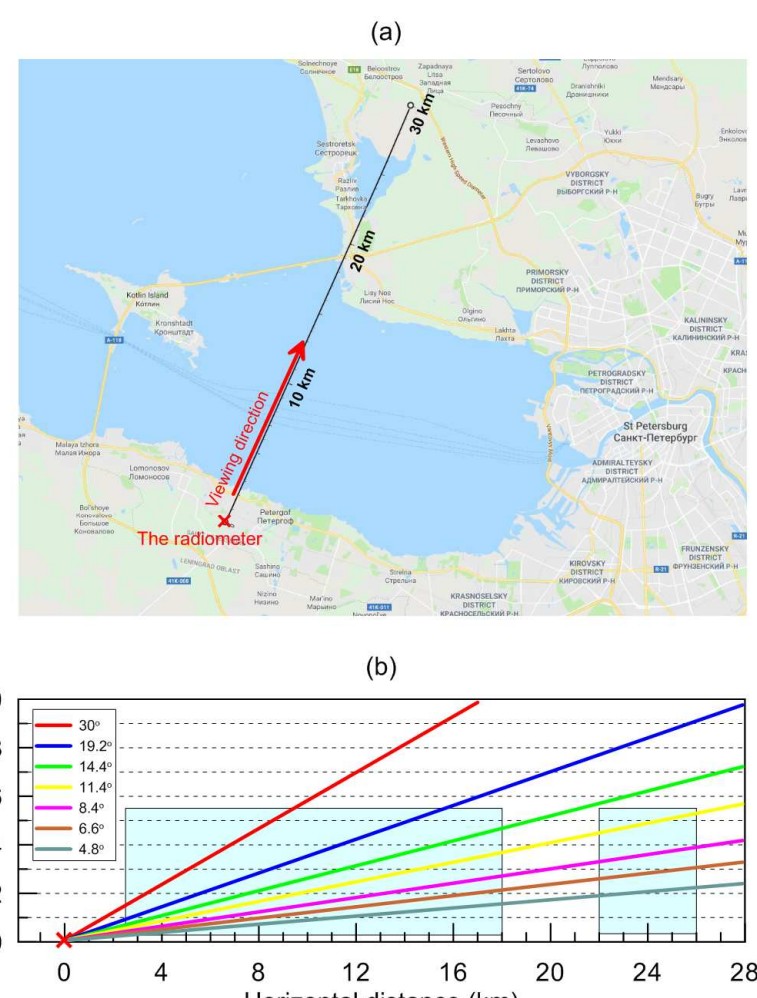

**Figure 1: (a) The location of the RPG-HATPRO radiometer and the viewing direction in the angular scanning mode. The black straight line is the distance scale. Map data © Google Maps 2019. (b) The viewing geometry in the vertical plane. Position of the radiometer is marked by the red cross. Colour lines represent the lines of sight for different elevation angles (see the legend). Blue boxes designate the atmospheric layer 0.3-5.5 km over water areas (see text). These figures are borrowed from the paper by Kostsov et al. (2020).**



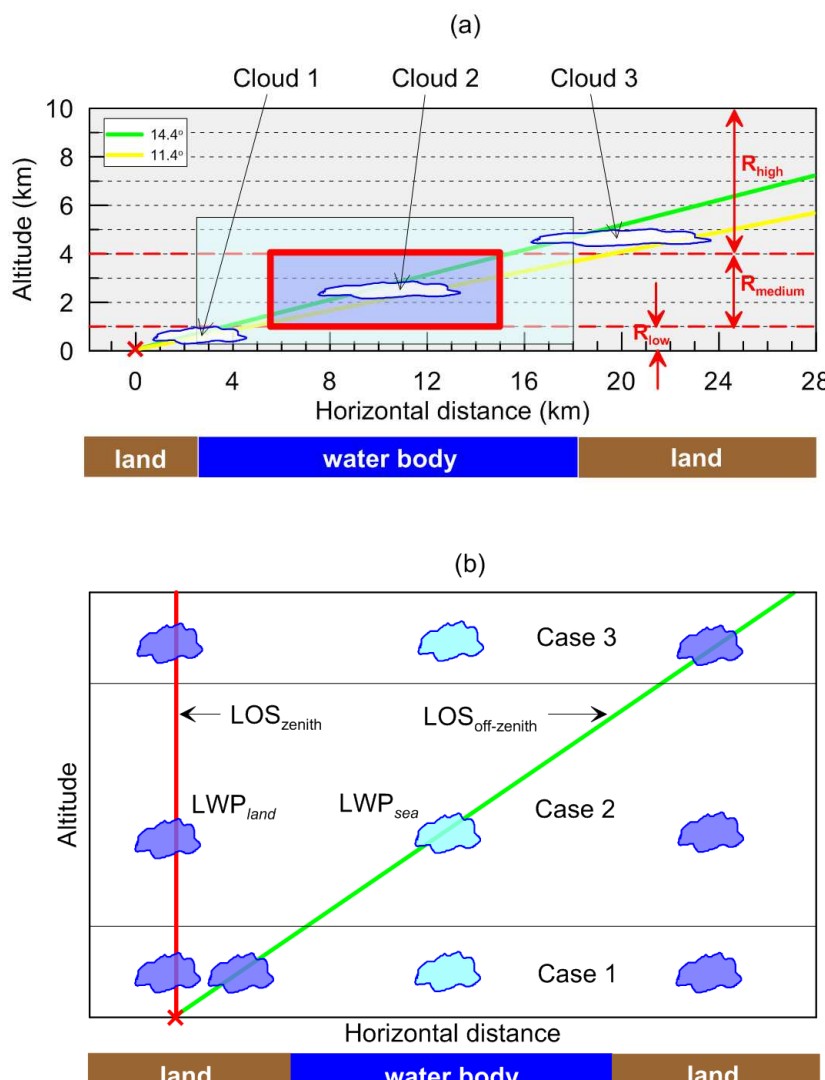

**Figure 2: (a)** Altitude ranges in the atmospheric model ($R_{low}$, $R_{medium}$, $R_{high}$) and the lines-of-sight which correspond to optimal retrieval strategy (the elevation angles are given in the legend); **(b)** Schematic picture of different cases of cloud location as an illustration to the discussion about the possibility to detect the LWP land-sea contrast. Clouds over land are marked by deep blue, clouds over sea – by light blue, the corresponding liquid water paths are designated as $LWP_{land}$ and $LWP_{sea}$. LOS is the line-of-sight, the radiometer position is marked by red cross. The scales of axes are arbitrary for illustrative purpose.

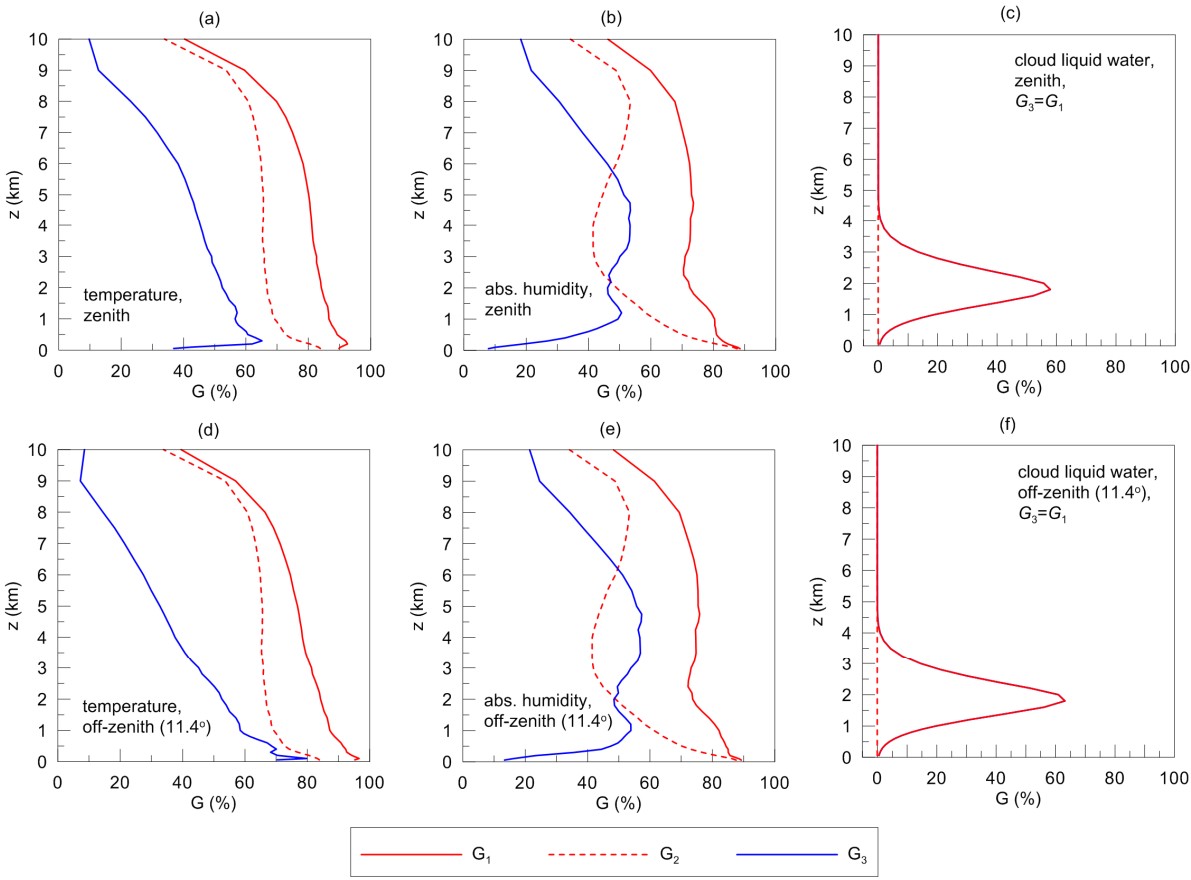

**Figure 3: The information gain functions $G(z)$ calculated for three parameters (temperature, absolute humidity, cloud liquid water - left, central and right panels in a row correspondingly) and for two elevation angles: 90° (zenith direction, upper row) and 11.4° (off-zenith direction, lower row).**

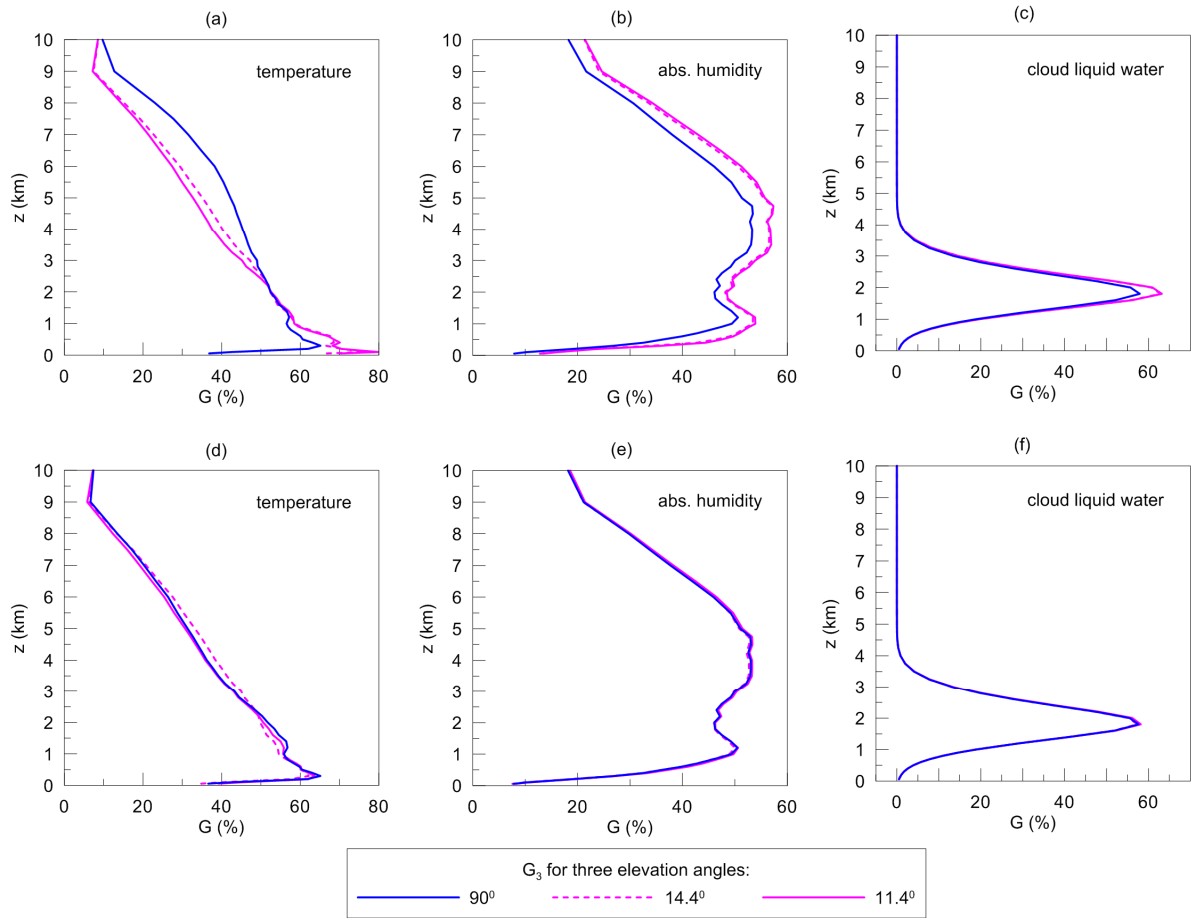

**Figure 4: The information gain function** $G_3(z)$ **calculated for three parameters (temperature, absolute humidity, cloud liquid water - left, central and right panels in a row correspondingly) and for three elevation angles (see the legend). Upper row: the original retrieval setup, lower row: the new setup presented in Table 2.**





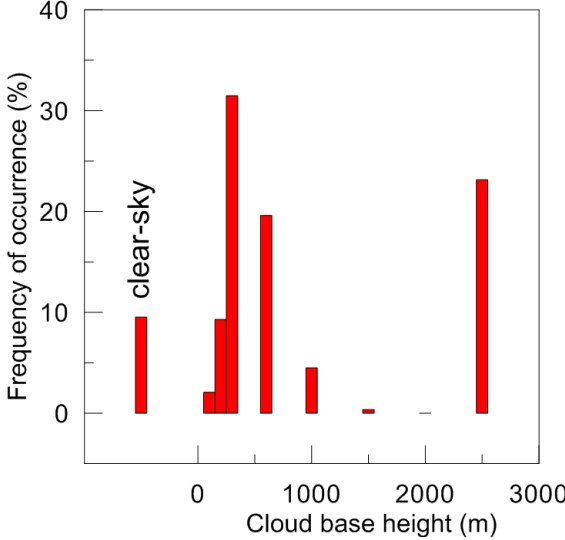

**Figure 5: The frequency of occurrence of clouds with different base heights, derived from the data of meteorological observations at the station in St.Petersburg in 2011-2017.**

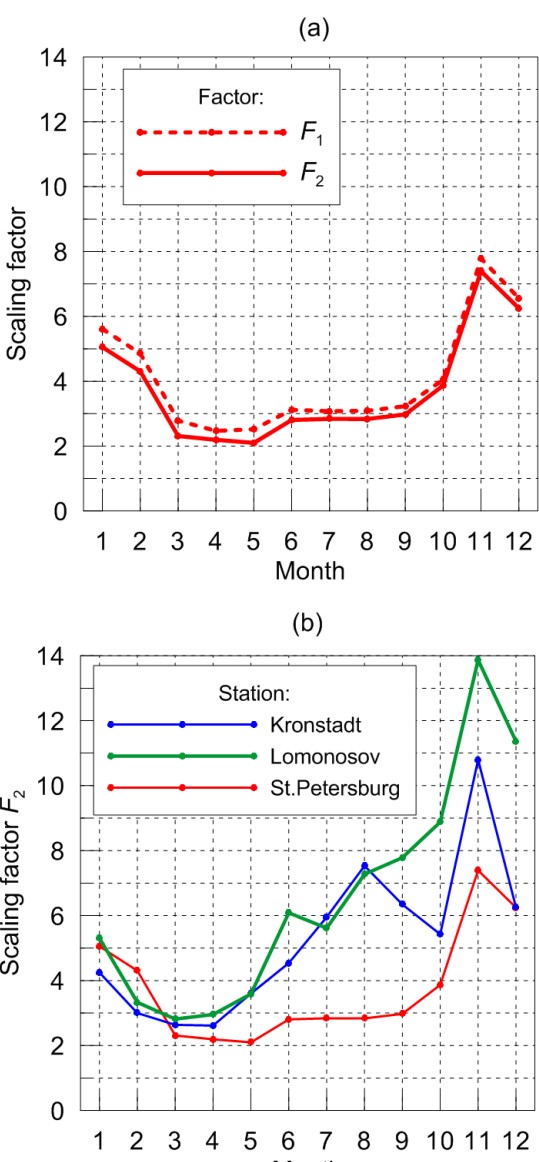

**Figure 6: (a)** Scaling factors $F_1$ and $F_2$ derived from the data of meteorological observations of the cloud base height at the St.Petersburg station in 2011-2017. **(b)** Scaling factor $F_2$ derived from the data of meteorological observations of the cloud base height at three stations in the vicinity of the radiometer (see the legend) in 2011-2017.







Figure 7: The values of the LWP retrieval bias $b_i$ for zenith and off-zenith geometry derived for summer months in 2013-2021. Index i designates the elevation angle (1=90°, 2=14.4°, 3=11.4°).




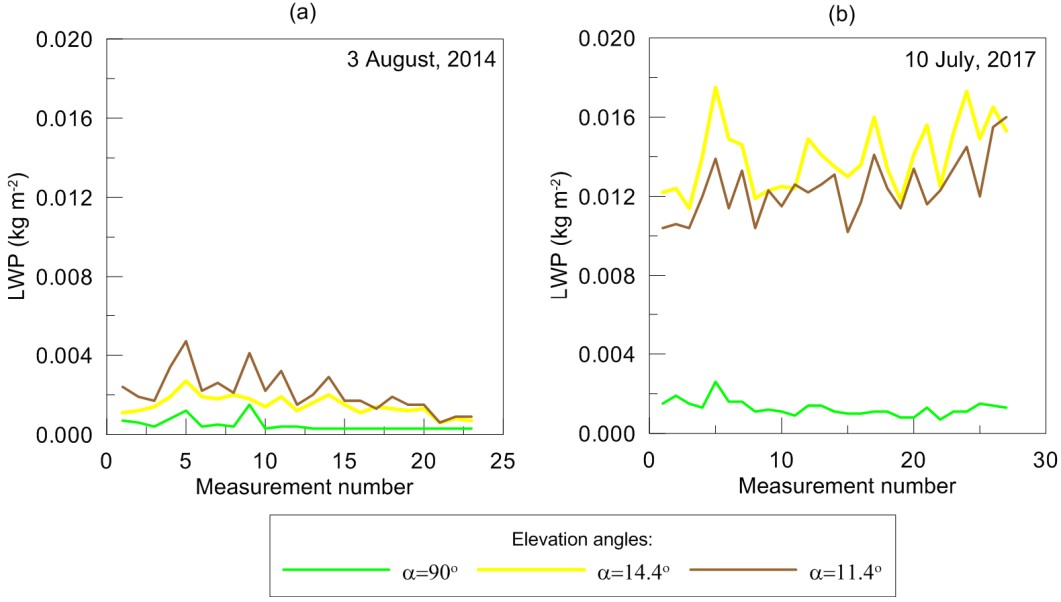

**Figure 8: The values of LWP retrieved from zenith and off-zenith geometry observations under perfect clear sky conditions (during the days of FTIR observations at the NDACC station at St.Petersburg State University).**





**Figure 9:** The results of the retrieval of LWP land-sea difference for summer months within the period 2013-2021. No scaling factors are applied.





**Figure 10: The estimations of the LWP land-sea contrast obtained for summer months within the period 2013-2021. The following scaling factors are applied: 5.3, 5.8, and 7.4 for June, July, and August respectively.**



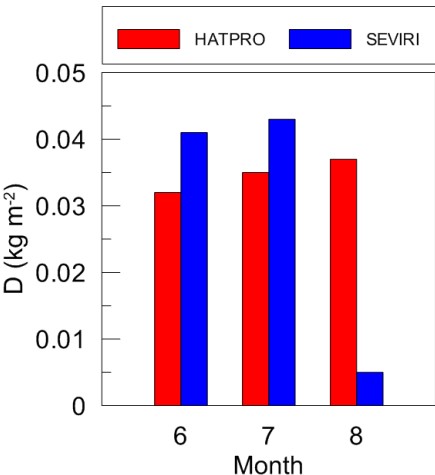

**Figure 11: The estimations of the LWP land-sea contrast in the region of Neva Bay of the Gulf of Finland for summer months. The results are shown which were derived from ground-based microwave measurements by the RPG-HATPRO instrument and from space-borne observations by the SEVIRI instrument during multi year periods.**