# Peer review of "Retrieval of the land-sea contrast of cloud liquid water path by applying a physical inversion algorithm to combined zenith and off-zenith ground-based microwave measurements"

_Atmospheric Measurement Techniques, 2021_

## Referee Comment (RC2)

Review for **"Retrieval of the land-sea contrast of cloud liquid water path by applying a physical inversion algorithm to combined zenith and off-zenith ground-based microwave measurements"**, submitted to Atmospheric Measurement Techniques

**Synthesis:**

This study presents a method that can be applied to microwave radiometers operated close to a water surface to determine differences in cloud liquid water path over land and water. The results of the newly developed algorithm for the ground-based microwave radiometer is compared to satellite observations.

In the current form, I do not recommend the study to be published. There are several points that should be addressed, in particular a thorough uncertainty analysis. Please find my comments below.

**General comments:**

- The study lacks a thorough uncertainty analysis, from the brightness temperature uncertainties to the retrieval errors. You discuss the error sources in Chapter 3.2, but you do not provide any values for the different errors which would be crucial for interpreting the results. The LWP differences in your study on the order of much less than 10 g/m² are well within the error ranges (both bias and random errors). Please provide a detailed uncertainty analysis including error bars in Figures 6, 7, 9, 10, and 11 !

- For your cloud retrieval you set up a so-called "cloud area of interest" which is between 1 and 4 km above ground. In general, there are many shallow clouds over the sea with both base and top below 1 km which would be completely neglected by this study. This is also confirmed by the cloud base observations used in the study (see line 481). It would be important to consider using lower elevation angles for the microwave radiometer to also catch the lower clouds. Otherwise, too many clouds are just not sampled for this study and a meaningful comparison with satellites becomes nearly impossible.

- The setup of the study lacks some comparison to land areas for the low elevation angles. It would have been good to also perform scans to the other direction (i.e. south-west) to check if the algorithm performs well over land, too. With a HATPRO microwave radiometer these bi-lateral scans can be performed easily. This would have provided a reference area to see whether the results between zenith and slant observations are in fact caused by the different surface conditions.

- The assumptions for the physical algorithm are very rough and the information on the forward model is quite sparse. The a-priori profiles (page 6, lines 191-192) are not described. Where do you get these profiles from, especially also the information on cloud liquid water? Which assumptions are used concerning clouds?  And what are the absorption models used in the study?

- I'm not convinced by the definition of the scaling factors $F_1$ and $F_2$. You are assuming that the vertical distribution of clouds is the same over all the years. Other factors like air or water temperature anomalies, as well as the precipitation patterns (dry/wet months) will strongly influence the cloud distribution of a special month which makes it very difficult to believe

that these scaling factors are stable for a special month. Did you see inter-annual differences for the period of the cloud base height dataset? If so, this would add uncertainty to the scaling factors. Furthermore, in Figure 6 it can be seen that the scaling factor depends very much on the location where the cloud base height has been taken. Also, it can be seen that there is quite a strong monthly variability. Do you have an explanation for this behaviour?

- Did you compare your LWP from the physical retrieval with statistical approaches? I would be interested to see whether there are differences, as you provided a new (physical) algorithm here.

- For the LWP bias correction (Figure 7), why are there sometimes considerable differences between zenith and off-zenith observations, and sometimes not (e.g. 2020 in June vs. July?). Are you sure that your bias correction algorithm is working properly? What might be the reason for these differences?

**Minor comments:**

- Page 2, lines 40-46: Too much detailed information
- Page 5, line 125: I would use the term "approach" instead of "algorithm"
- Page 5, line 145: What do you mean by "model form"?
- Page 12, lines 315-316: Why do you think that the LWP contrast is the same for cloud scenes with different cloud base heights? I don't think that is a valid assumption!
- Page 17, line 452: I don't understand what you want to say here. Do you mean that the sensitivity is instrument specific? If so, then please write it that way.

---

## Author Comment (AC1)

**The reply to the anonymous referee #1 (RC1)**

We are thankful to the referee for the detailed analysis of our study and for the constructive criticism. We agree with most of statements made by the referee and we took into account almost all of them while revising our paper.

Below, the actual comments of the referee are given in **`bold courier font and blue colour`**. The text added to the revised version of the manuscript is marked by red colour.

*Notice1: Since both anonymous referees made several similar remarks, our answers to these remarks which are given in both replies are identical.*

*Notice2: Numbering of figures and sections has been changed considerably in the revised version of the manuscript.*

**`Summary`**

**`In this paper microwave radiometer (MWR) measurements at a coastal location in Northern Europe are utilized to derive liquid water path (LWP) information using a physical retrieval approach for the cloud liquid water content (LWC). Differences in zenith and off-zentih observations are attributed to a land-sea contrast and compared to satellite data for a multi-year period.`**

**`General Comments`**

**`The authors dedicate large parts of the paper to the description of a physical retrieval and a scaling approach based on the theoretical positioning of clouds relative to their measurement geometry to account for different cloud base heights (CBH). This aims towards an improvement (compared to previous studies) of the complex task to accurately quantify spatial cloud liquid water contrasts using a single MWR.`**

The formulation of the essence of our study made by the referee is absolutely exact. And indeed, the task appeared to be rather complex. The present study is the continuation of our previous research and aims to improve the approach to quantification of the LWP land-sea contrast and, as a consequence, to increase the accuracy.

**`Despite this effort, the methods and assumptions applied here still encompass large uncertainties, which makes it difficult to interpret the highly variable quantity of interest as described below.`**

We agree with this statement to a certain extent. We admit that the obtained results are not yet perfect and the uncertainties exist but at the same time we believe that we made considerable progress compared to our previous work. We hope that the obtained results can be interesting for the scientific community both in term of capabilities of the ground-based microwave observations and in term of the LWP land-sea contrast features.

**`In the motivation it is mentioned that previous findings of a positive trend in the land-sea LWP contrast and a diurnal cycle during June and July need further research (l. 95). These points are not addressed further and no explanation is given on why this trend is not evident in the current study.`**

In the revised version in the Subsection 5.3 we address these points as follows:

**5.3 Trend assessment and the problem of analysis of diurnal evolution of the LWP land-sea contrast**

Space-borne observations by the SEVIRI instrument have revealed a positive multi-year (2011-2017) trend in the LWP land-sea contrast which appeared to be statistically significant for four locations in Northern Europe (https://doi.org/10.5194/acp-2021-387-AC3, last access 12 April 2022). For cold season, the statistically significant trend is observed for Lakes Onega and Ilmen. For warm season, the significant trend is observed for Gulf of Riga of the Baltic Sea and for the Neva River Bay. The Neva River Bay is just the water body for which the LWP land-sea contrast in summer is investigated in the present study. We made the attempt to check whether this trend shows up in our ground-based data. Fig. 9 demonstrates the inter-annual variability of the LWP contrast values $D_1$ and $D_2$ averaged over the entire summer period (June, July and August taken together). Also, the inter-annual variability of the LWP values over land (LWP$_{zen}$) is shown together with the number of measurements used for averaging. One can see that no trend can be detected. The coefficient of determination (R-squared) for the LWP contrast is rather low and equal to 0.215. Both the LWP over land and the LWP contrast are highly variable. One of the reasons for that is the very large variability of the number of measurements used for averaging. The smallest number 350 was in 2016 and the largest number 2061 was in 2019. This number explicitly depends on the quality control criteria and implicitly depends on the weather conditions (non-rainy cases only were allowed) and on the operational state of the instrument. One should also not forget about the upper and lower limits of the altitude region which is remotely sensed over water in the current experiment (1-4 km according to applied cloud model). So, the only conclusion that we can make is that measurements of HATPRO during 2013-2021 provided no evidence of any multi-year trend of the LWP land-sea contrast for clouds with CBH in the range 1-4 km.

The diurnal evolution of the LWP land-sea contrast is another interesting feature which has been revealed from the space-borne measurements by the SEVIRI instrument (Kostsov, 2021). In the present study we do not analyse the diurnal cycle since we consider current experimental setup as not suitable for solving the problem of studying diurnal cycle of the land-sea contrast. There are two main reasons for that. First, the altitude area for remote sensing over water is very limited and does not include altitudes below 1 km where clouds appear most often according to local statistics. And second, the remote measurements over water are not frequent enough to obtain information on the LWP contrast evolution during a single day. Every measurement is a kind of a snapshot of an instantaneous cloud scene which is to a certain extent random. This circumstance distinguishes ground-based observations from space-borne observations since every single measurement from satellite provides an average estimate of LWP due to sampling over finite area (pixel).

[Figure]

**Figure 9: Inter-annual variability of the LWP land-sea contrast values ($D_1$ and $D_2$) and the LWP over land (LWP$_{zen}$) in the region of Neva Bay of the Gulf of Finland for summer period as derived from ground-based microwave measurements by the RPG-HATPRO instrument. Blue symbols show the number of measurements during each summer period which passed quality control and were used for statistics.**

We also added two points to conclusions:

4) Measurements of HATPRO during 2013-2021 provided no evidence of any multi-year trend of the LWP land-sea contrast in summer for clouds with cloud base height in the range 1-4 km and demonstrated high

inter-annual variability of the LWP contrast. It should be noted that number of measurements which passed the quality control and were used for averaging varied considerably from year to year.

5) In the present study the diurnal cycle of the LWP land-sea contrast was not analysed since we consider current experimental setup as not suitable for solving this problem due to limited altitude range of the air portion which is remotely sensed over water and due to insufficient frequency of angular scans.

**Furthermore the high variability of bias values and LWP differences visible in Fig. 7, 9, 10 is not explained. Discussing the influence of different synoptic situations or even deriving an IWV retrieval (zenith only) would add more value to the paper.**

We agree with this remark and we added the discussion of the variability of bias values in Section 4:

We tried to find the reasons for the variability of bias values for off-zenith geometry. The primary guess was the influence of different synoptic situations, namely the situations with different integrated water vapour (IWV). However, this guess was not confirmed. As shown in Fig. 7a, there is no noticeable correlation between the monthly mean values of bias $b_2$ and monthly mean values of IWV during the period 2013-2021. The important notice should be made: we calculated mean IWV values exactly for the time periods which were selected for bias assessment. In Fig. 7a one can notice large inter-annual variability of IWV for all summer months. Lowest mean IWV values are detected in June, and the highest values are detected in July. We made an attempt to find a correlation between monthly mean values of bias $b_2$ and monthly mean values of temperature at the ground level during the period 2013-2021, but no noticeable correlation has been found. As a result, the conclusion was made that the influence of synoptic situation on LWP bias was not detected.

Another idea about the reason for off-zenith bias variations was related to possible horizontal inhomogeneities of temperature in the vicinity of the radiometer. It has been already noted that the radiometer is installed on the roof of the building which is heated by solar irradiance. The building is about 200 m long and the line-of-sight passes directly over it. Fig. 7b shows the values of the LWP retrieval bias $b_2$ for summer months in 2013-2021 as a function of mean value of the temperature horizontal difference $\Delta T$ at 100 m altitude near the radiometer for these months. This temperature difference was derived from the temperature profiles which were retrieved using zenith and off-zenith geometry. Of course, one should keep in mind that the spatial resolution is very poor, but nevertheless this temperature difference can be used either an indicator of horizontal temperature gradients or an indicator of some effects which interfere in the retrieval process. The assessment of correlations between $b_2$ and $\Delta T$ using the Fisher criterion for small number of samples has shown that correlations in July and August are statistically significant. The correlation is also statistically significant if the entire summer period is analysed without division by months. So we accepted the hypothesis that the reason for the variability of LWP bias for off-zenith observations can be a temperature horizontal inhomogeneity in the close vicinity of the radiometer. Probable mechanism could be the following: under conditions of considerable temperature inhomogeneity, the temperature error propagates into the LWP error and the LWP bias increases.

[Figure]

**Figure 7: (a) The values of the LWP retrieval bias $b_2$ for summer months in 2013-2021 as a function of mean value of IWV for these months. (b) The values of the LWP retrieval bias $b_2$ for summer months in 2013-2021 as a function of mean value of the temperature horizontal difference at 100 m altitude near the radiometer for these months. Dashed lines demonstrate linear fits. The values of correlation coefficient are given as $r$.**

We made the analysis of the variations of the LWP contrast using the same approach as applied to LWP bias variations. Corresponding discussion has been added to Section 6:

> We made the analysis of the variations of the LWP contrast using the same approach as applied to LWP bias variations (please see Section 5). No significant correlations with the integrated water vapour, temperature at the ground level and LWP over land (over the radiometer) were detected. As a result, the obvious conclusion was made that the comprehensive analysis of the LWP contrast requires the data on the synoptic situation not only at the location of the radiometer but also over the Neva River Bay.

**For the presented discussion and conclusions drawn here Fig. 11 would be sufficient, but a general agreement with SEVIRI data (except for August) has been shown already in Kostov et al. (2020). The LWP differences (not scaled) of the physical retrieval seem to be in the same range as the ones retrieved from a regression retrieval (comparing Fig. 9 from the present study and Fig. 18 from Kostov et al. (2020)).**

The regression and physical approaches are used for processing microwave measurements for decades and in many cases the accuracies of these approaches are comparable. However, it is widely admitted that the physical approach has certain advantages. The problem which is solved in the present study brings forward very strong requirements to the accuracy, therefore the application of the physical approach seems to be a logical step. The disadvantages of the regression approach were enumerated in the first paragraph of Section 2 with the reference to our previous work which was specially focused on the comparison of physical and regression algorithms. In the original version of the manuscript, we deliberately did not present the comparison of the results obtained by physical approach with the results obtained by regression approach because it was not the main focus of the work. Since the referee attracted attention to this issue, we clarified it in the revised version in Section 5.4 as follows:

**5.4 Cross-comparison of the LWP contrast values derived by the physical algorithm and by the regression algorithm as a means of validation of the obtained results**

We would like to discuss briefly the problem of choice of a retrieval algorithm (physical or regression) for the specific task which we solve in this study and the problem of validation of obtained results. The LWP contrast values for 2013 and 2014 obtained by the regression algorithm in the previous study (Kostsov et al., 2020 and Fig. 18 therein) are of the same range as ones produced by the physical approach in the present work. As one can see in Fig. 6, the years 2013 and 2014 are somewhat special: the LWP retrieval bias is minimal and comparable for zenith and off-zenith observations. It is no wonder that both algorithm work well. The situation is completely different for the years 2015-2021 when the LWP retrieval bias values for off-zenith observations are considerably larger. In order to compare the results produced by the two algorithms we made two tests. In the first test, we applied the regression algorithm to the HATPRO measurements which successfully passed quality control (convergence of the physical retrieval process and spectral residual check). In the second test we applied regression algorithm to all measurements (no quality control at all). It is important to emphasise that in both cases the bias of the LWP retrievals by regression method was estimated from measurements selected by the physical method. The outcome was quite demonstrative. In the first case, the results produced by the physical and regression algorithms were in good qualitative and quantitative agreement. They are demonstrated in Fig. 10, panels (a), (c), and (e). One can see the overall very good agreement with only two noticeable deviations of the results obtained by regression algorithm from the results of the physical algorithm (July 2016, August 2013). In the second test, the results obtained by the two algorithms were completely different and they are not shown. These tests clearly indicated the superiority of the physical algorithm in the problem of assessment of the LWP contrast. As it was noted in the beginning of Section 2, the identification of cloud-free periods of time and quality checks are impossible using the results obtained by the regression method. Therefore, it is necessary to emphasise once again that for the data set of HATPRO measurements, which we process in the present study, the application of the regression algorithm only would have produced wrong results.

Comparison of the results obtained by different algorithms can be a valuable means of cross-validation. A vivid demonstration of such cross-validation has been presented in the study by Kostsov et al. (2018a). In this study

a special case of ground-based measurements by HATPRO radiometer after a rain event was considered. After rain, the radome of microwave instrument is wet for some time. During this period of time measurements are erroneous. Kostsov et al. (2018a) have shown that the discrepancy between the outputs of regression and physical algorithms is maximal during a rain and becomes smaller and smaller after a rain event while water evaporates from the radome. This effect clearly demonstrates that controlling the discrepancy between the outputs of regression and physical algorithms can be a means of validation of the final results. This statement refers not only to an "after a rain" period but has a general character and can be applied to different sophisticated situations and problems like the problem of detecting the LWP land-sea contrast. That is why, one can consider the good agreement of LWP contrast values produced by regression and physical algorithms as a successful validation of the obtained results.

In order to get one more confirmation of the validity of the obtained results, we processed the HATPRO data applying the "standard" atmospheric model which is used in routine zenith microwave observations and is characterized by the cloud altitude range 0.3-5.5 km. The data processing procedure was the same as for the model with cloud altitude range 1-4 km, with two exceptions:
- the threshold for bias assessment was taken as 0.015 kg m-2 since the "standard" model has larger a priori uncertainty of cloud liquid water and hence larger bias;
- the retrieval setup was not specially tuned to provide equal sensitivity of zenith and off-zenith measurements to atmospheric parameters, the retrieval setup for the model with cloud altitude range 1-4 km was used without any modifications.

The results of the derivation of the LWP land-sea contrast with the "standard" model are presented in Fig. 10, panels (b), (d), and (f). One can see that the discrepancies between the outputs of physical and regression algorithms are noticeably larger than for the "1-4 km" model. Also, the differences between $D_1$ and $D_2$ values for both algorithms are very much larger than ones obtained for the "1-4 km" model. These two facts indicate that in case of "standard" model the results are very much less self consistent than for the "1-4 km" model. Comparison of left and right panels in Fig. 10 leads to the conclusion that the modification of "standard" model for the specific task of detecting the LWP land-sea contrast was the correct decision.

[Figure]

**Figure 10: Comparison of the LWP land-sea contrast values ($D_1$ and $D_2$) obtained by the physical algorithm and by the regression algorithm for June (a, b), July (c, d) and August (e, f) in the period 2013-2021 and for two cloud altitude range models: 1-4 km (a, c, e) and 0.3-5.5 km (b, d, f).**

The setup of the physical retrieval is not clear to me, in particular why LWC is retrieved (instaed of LWP) despite the low information content of a MWR to the vertical distribution of cloud liquid water and how the LWC prior information was derived without knowing the cloud boundaries.

The physics of the radiative transfer in the microwave region, in particular non-selective absorption of cloud droplets implies low information on LWC profile. In our opinion, it does not matter how to formulate mathematically the inverse problem, the outcome will be the same: we derive only a LWP and not a LWC profile. The physical retrieval, as we formulate it, is based on the linearization and inversion of the radiative transfer equation. It is just very convenient to use LWC profile along with profiles of all other parameters which influence the microwave radiative transfer. Low sensitivity to vertical distribution of LWC helps to avoid knowing cloud boundaries. A priori profile of LWC is assumed as nearly zero ($10^{-8}$ kg m$^{-2}$) in a large altitude interval (1-4 km) which simulates cloud-free conditions. In contrast, the variability of LWC is quite large which allows retrieving high range of LWP. For clarification, we added this information in the end of subsection 3.1:

A priori profile of LWC is assumed as nearly zero profile ($10$-$8$ kg m$^{-2}$) in the altitude interval (1-4 km) which simulates cloud-free conditions. In contrast, the variability of LWC is quite large which allows retrieving high range of LWP. For temperature and absolute humidity a priori profiles, we use the data from nearest radiosonde station which were averaged over several years of radiosonde launches.

An error related to the misplacement of clouds is mentioned, but assumed to be the same for zenith and off-zenith observations (l. 361). Also the cloud altitude range has been modified to 1-4 km for the zenith retrieval. To my knowledge this makes it inconsistent with other prior information and could introduce a bias in the resulting LWP contrast, especially regarding the fact that most observed CBH are below 1 km (Fig. 5). A discussion on the retrieval error should be added in order put the LWP differences into context.

We can not agree that "this makes it inconsistent with other prior information". The cloud altitude range is not bound to any other a priori information. As it has been emphasised in the text, we organised our atmospheric model in such a way that it exactly corresponds to the problem of investigation of the LWP land-sea contrast. In other words, according to the model, non-zenith observations give the information on the LWP over water only, not over land. It is an advantage of the model since the absorption is calculated for proper temperatures for the clouds over water. We can agree that this model is idealised. Therefore, while preparing the revised version of the manuscript, we decided to make calculations with the "standard" cloud altitude range of 0.3-5.5 km and to compare the results. We included the results of the comparison in the revised version in Section 5.4:

In order to get one more confirmation of the validity of the obtained results, we processed the HATPRO data applying the "standard" atmospheric model which is used in routine zenith microwave observations and is characterized by the cloud altitude range 0.3-5.5 km. The data processing procedure was the same as for the model with cloud altitude range 1-4 km, with two exceptions:
- the threshold for bias assessment was taken as 0.015 kg m-2 since the "standard" model has larger a priori uncertainty of cloud liquid water and hence larger bias;
- the retrieval setup was not specially tuned to provide equal sensitivity of zenith and off-zenith measurements to atmospheric parameters, the retrieval setup for the model with cloud altitude range 1-4 km was used without any modifications.

The results of the derivation of the LWP land-sea contrast with the "standard" model are presented in Fig. 10, panels (b), (d), and (f). One can see that the discrepancies between the outputs of physical and regression algorithms are noticeably larger than for the "1-4 km" model. Also, the differences between $D_1$ and $D_2$ values for both algorithms

are very much larger than ones obtained for the "1-4 km" model. These two facts indicate that in case of "standard" model the results are very much less self consistent than for the "1-4 km" model. Comparison of left and right panels in Fig. 10 leads to the conclusion that the modification of "standard" model for the specific task of detecting the LWP land-sea contrast was the correct decision.

Notice: mentioned Fig. 10 is already shown above.

**It is mentioned that information about temperature in liquid water cloud layers is an advantage of the physical retrieval (l. 131), but the rather smooth MWR profiles do not accurately account for that.**

Certainly, we agree that the smoothing is very large. But still, a retrieved profile is better than an a priori profile. Besides, the retrieved profile is required for the quality check which is based on the calculation of spectral residual. Such check is impossible using a priori profile.

**Suggestions for a further improvement of the retrieval performance include using nearby radiosonde data, or the LWP from a regression retrieval as a first guess. Since the focus is on the summer period, the retrieval setup could be done only using prior information from that season.**

We are thankful to the referee for these suggestions, but a first guess specific for certain moment of time is not necessary. Under typical conditions with no temperature or humidity inversions, when the radome of radiometer is not wet after rain or snow, the iterative retrieval process converges after 4-5 iterations. It is fast enough. And we have an opposite opinion on the problem whether to choose seasonal a priori information or averaged over a year. We use temperature and absolute humidity profiles from the nearest radiosonde station averaged over several years as a priori profiles of these parameters. By the way, the same approach with multi-year statistics (not seasonal) is used by the manufacturer of the HATPRO radiometer for creating the built-in regression algorithm. As mentioned above, a priori profile of LWC is nearly zero ($10^{-8}$ kg m$^{-2}$) which simulates cloud-free conditions. In contrast, the variability of LWC is quite large which allows retrieving high range of LWP.

**The 19.2° elevation angle was removed from the analysis, although it still probes a significant amount of the designated area of interest. According to the hypothetical cloud scheme it would also contain information on high clouds over water and could be compared to lower elevation angles. One hypothesis could be that the contrast remains similar if high clouds are decoupled from the underlying land-sea contrast. Also, the assumption that the contrast for low and high clouds is zero (l. 293) and that the true value is always larger does not hold for the case of advected clouds. For the described case 1 for example, low level and relatively low LWP clouds advected over from the water into the off-zenith observations would cause a LWP contrast.**

The authors are thankful to the referee for this insightful comment and the idea about low clouds advected from water to land which can cause the LWP contrast. In our opinion, this interesting case and the case with high clouds mentioned by the esteemed referee need thorough consideration which is possible only after all questions relevant to the retrieval scheme are solved. Therefore we leave this consideration beyond the scope of the present study. Concerning the question whether to remove or keep the 19.2° elevation angle, we decided that the better way to give an answer would be to make calculations for this angle. The results are described in the revised version in Section 5.5:

**5.5 Test retrievals for elevation angles 19.2° and 8.4°**

It is interesting to investigate the dependence of the derived values of the LWP land-sea contrast on the elevation angle. It has been explained above that two elevation angles 11.4° and 14.4° have been selected as optimal for solving the task of detecting the LWP land-sea contrast for specific geometry of the experiment which is determined

by the location of the instrument and the size of the water body. The lines of sight for these two angles intersect the area of interest over water body spanning vertically from 1 to 4 km. Despite the fact that there can be possible influence of the underlying surface on the results of microwave measurements when elevation angles are smaller than 10°, test retrievals were made for the elevation angle 8.4°. Also, test retrievals were made for the elevation angle 19.2° in order to obtain information which could be sufficient for derivation of a dependence on elevation angle. Both "extra" elevation angles fit well the atmospheric model compiled for the LWP contrast retrievals since the corresponding measurements still probe a significant amount of the designated area of interest (see Fig. 1).

Validation of the results for two extra elevation angles was made on the basis of comparison of the LWP contrast values derived by physical and regression algorithms. This comparison has shown that the discrepancy between the outputs of physical and regression algorithms for the angle 19.2° has the same magnitude as for the optimal angles 11.4° and 14.4°. For the angle 8.4°, the discrepancy is noticeably larger. This fact is the confirmation of the recommendation to avoid using elevation angles smaller than 10°. Nevertheless, we kept all results for analysis.

The values of the LWP land-sea contrast are plotted as a function of elevation angle in Fig. 11 for each summer month and each year separately. One can notice a clear and well pronounced dependence of the LWP contrast on elevation angle which is characteristic for all cases except two. Maximal contrasts are detected for the optimal elevation angles 11.4° and 14.4°. For "extra" angles, the contrasts are always lower (or sometimes equal to contrasts for optimal angles). Two exceptions refer to July 2014 and July 2016: in these cases the dependence on elevation angle is absent. The detected dependence can be explained by clear physical reasons. For large elevation angles, the portions of air, which are probed, are shifted up and towards the instrument. The liquid water content for high clouds usually is less than for the lower clouds, hence the contrasts should be smaller. Also, the contrast in the vicinity of the coastline is expected to be smaller. For small elevation angles, the air portion, which is probed, is shifted down and towards the opposite shore. The influence of clouds over the opposite shore can cause the decrease of mean values of the LWP contrast. One can see that in some cases the LWP contrast obtained for extra elevation angles is negative. These cases require special study, which should be based probably on the analysis of synoptic situations.

[Figure]

Figure 11: Retrieved LWP land-sea contrast as a function of elevation angle for June, July and August in 2013-2021. Symbols correspond to four discrete values of elevation angle (19.2°, 14.4°, 11.4°, and 8.4°) and are connected by lines only for illustrative purpose.

The scaling factors for the "true" LWP land-sea contrast are based on human observer CBH statistics, which was derived over several years and averaged data from three different stations are shown (Fig. 6). The factor represents the ratio of the number of all clouds to only medium height clouds (1-4 km), but it includes observed clouds at 2.5 km and above, since no more distinction is made. The applied assumptions make this scaling highly uncertain, while the general conclusion remains the same other than a better agreement to SEVIRI data. It would be good to know the standard deviation of this value and demonstrate the resulting uncertainty for the scaled LWP contrast alongside with an uncertainty estimate for the SEVIRI results.

In order to provide a kind of validation of scaling factors derived from statistics collected by human observers at meteorological stations we compared scaling factors from meteorological stations with the factors derived from ceilometer measurements. We also added the discussion of the uncertainty of scaling factors in Section 6.2:

In order to provide a kind of validation of scaling factors derived from meteorological data we calculated scaling factors using the CBH measurements made with a ceilometer. The CHM 15k ceilometer was in operation at the observational site of St.Petersburg University in 2013, 2014 and the first half of 2015. It was installed on the metal tower on the roof of the building just nearby the HATPRO microwave radiometer. The full description of the CHM 15k ceilometer can be found on the web page of a manufacturer (https://www.lufft.com/products/cloud-height-snow-depth-sensors-288/ceilometer-chm-15k-nimbus-2300/, last access 6 May 2022). The CBH values were derived by the original data processing algorithm embedded in the instrument. For the purpose of comparison, the selection of CBH data for calculation of scaling factors was done exactly in the same way as the selection of data provided by human observers at meteorological stations.

Resulting monthly values of $F_1$ and $F_2$ for daytime are shown in Fig. 13. Since scaling factors obtained from the ceilometer observations appeared to be very similar to factors at the meteorological station St.Petersburg, they are shown in one plot Fig. 13a. The factors $F_1$ and $F_2$ from ceilometer observations are practically indistinguishable, therefore only the $F_2$ factor is shown in this plot. Comparison of $F_1$ and $F_2$ obtained from the St.Petersburg station records is given in Fig. 13a and demonstrates also very similar values for all months. Minimum monthly values of scaling factors from all data sources are observed in spring and summer, maximal values – in autumn and winter. In spring, summer and early autumn, the differences between the multi-year average values of $F_2$ from meteorological station and the values of $F_2$ for specific years are the smallest if compared to late autumn and winter. This fact is the strong indication that the cloud statistics are stable for these months. It is surprising that for June and July the results obtained from ceilometer measurements in 2013, 2014 and 2015 are nearly identical.

Comparison of scale factor $F_2$ derived from the records of all three meteorological stations is shown in Fig. 13b. One can see that the main feature of intra-annual variability is the same for all stations: minimal values in spring and maximal in late autumn. Apparently, this behavior is due to the predominance of the lowest clouds in late autumn, and of the medium and high clouds – in spring. But there are also noticeable differences. First, while $F_2$ for the St.Petersburg station is nearly constant from March to September, $F_2$ for the Kronstadt and Lomonosov stations increases during this period. From December to April, the values of $F_2$ for all three stations are very similar. From May to November, there is noticeable difference between the values obtained at St.Peterburg station and two other stations. For Kronstadt and Lomonosov, the scaling factor is approximately 1.5-2.5 times higher than for St.Petersburg.

Comparison of Fig. 13a with Fig. 13b leads to several principal conclusions. First, the scaling factors are essentially different in summer and autumn for so-called "continental" and "marine" locations. Second, the location of the HATPRO radiometer can be attributed to the category of "continental" locations. And third, while for "continental" locations there is some evidence of the stability of scaling factors for specific months, we can not prove it for "marine" locations. As a result, the problem arises what scaling factor to choose and how to estimate the uncertainty of chosen factor. Since the main part of the line of sight of the radiometer passes over water body, it is reasonable to choose the scaling factor obtained at marine locations. The average scaling factors derived from the data obtained at the two meteorological stations Lomonosov and Kronstadt are 5.3, 5.8, and 7.4 for June, July, and August respectively. These values have been chosen for scaling the LWP land-sea contrast data (see Section 6 below). The half difference between these values and the correspondent values from the "continental" station St.Petersburg seems to be reasonable uncertainty estimate. So finally we have the following data for $F_2$: 5.3±1.3 for June, 5.8±1.5 for July and 7.4±2.3 for August.

[Figure]

**Figure 13: (a) Scaling factors $F_1$ and $F_2$ derived from the data of meteorological observations of the cloud base height at the St.Petersburg station in 2011-2017 (average values) and from ceilometer observations in 2013, 2014 and 2015 at the location of the HATPRO radiometer. (b) Scaling factor $F_2$ derived from the data of meteorological observations of the cloud base height at three stations in the vicinity of the radiometer (see the legend) in 2011-2017 (average values).**

**The LWP bias assessment is made using a threshold of 5 g/m² for identifying liquid water cloud free cases ("clear sky" might not be the right term). How does this value compare to the corresponding retrieval uncertainty? If it is chosen too low the bias would be underestimated. Although it is not possible for off-zenith observations due to the low temporal resolution, the method of using the LWP standard deviation from zenith observations could be used as an additional criterion to identify liquid water cloud free cases. Accurate estimates of the bias for zenith and off-zenith observations is important for the assumption that there is no contrast in the case of liquid water cloud free cases (l. 289).**

We agree with the remark of the referee about a threshold. We added the text and the new figure in Section 4 to show that our choice of the threshold had been correct:

The question what threshold should be selected for identifying cloud free cases for zenith observations needs some explanations. If it is chosen too low and the random errors of retrievals are too high the bias would be underestimated. In order to illustrate the problem of choosing the threshold, we present Fig. 5 where the random

error of LWP retrieval is shown as a function of retrieved LWP for low values of LWP. The random error of LWP retrieval is unique for each single measurement. It is derived from Fisher matrix (see Eq. 9) and Fisher matrix is calculated using retrieved profiles rather than a priori profiles. Measurements during July 2014 are taken as an example. Two models of cloud altitude range are considered: the model with the 1-4 km range, which is the main model of the present study, and the model with the 0.3-5.5 km range which is the standard model for routine zenith observations. Thresholds for cloud free cases have been selected as 0.005 kg m$^{-2}$ for main model and 0.015 kg m$^{-2}$ for standard model. The "standard" model is characterised by larger retrieval errors since the expanded altitude range implies larger a priori uncertainty of the LWC profile in the lower layers. Obviously, the retrieved LWP values for "standard" model are larger also. One can see that the areas of high density of retrieved LWP are well below the threshold lines. Taking into account the average values of retrieval errors (about 0.005 kg m$^{-2}$ for main model and about 0.007 kg m$^{-2}$ for "standard" model) one can conclude that most of selected measurements are within the range defined by thresholds for both models and that the bias values will not be underestimated.

[Figure]

**Figure 5: Random error of LWP retrieval as a function of LWP value for low LWP. Measurements during July 2014 are taken as an example. The results corresponding to two models of cloud altitude range are shown by red and blue colors (see the legend). Vertical color lines indicate thresholds chosen for cloud free scenes.**

**Specific Comments**

**Figure 1: a reference to Kostov et al. (2020) might be sufficient here and this figure could be removed**

We would like to keep this figure because, first, it helps a reader to understand the experimental setup without consulting previous works, and, second, because in the revised version we present the results for two more elevation angles. Fig. 1b vividly demonstrates the optical lines of sight for the entire set of elevation angles. Also, in the revised version we indicated the location of meteorological stations in Fig. 1a.

**Section 1.1: Referencing of different scanning radiometers could be shortened. Instead literature related to the topic of land-sea contrasts should be presented.**

We removed completely the description of different scanning radiometers and instead provided brief description of the necessity to study land-sea contrasts. We would like to note that according to classification of "Atmospheric Measurement Techniques" the topic of the article is "Data Processing and Information Retrieval". Therefore we believe that review of literature about physical mechanisms driving land-sea contrasts is beyond the scope of the study. The revised text in the beginning of the Introduction section is the following now:

Information on the land-sea contrasts of different atmospheric parameters is required for solving a wide range of problems relevant to climate change, interactions of the atmosphere with underlying surface, and validation of space-borne remote measurements of atmospheric state and composition. For example, in the climate change studies, it was shown that surface temperature over land increases more rapidly than over sea in response to greenhouse gas forcing (see Dong et al., 2009 and references therein). Cloud formation is one of multiple processes involved in this effect along with moisture transport from sea to land (Joshi et al., 2008). Land-ocean contrasts and diurnal cycles over land and ocean of upper tropospheric humidity were studied by Chung et al. (2013) on the basis of reanalysis data sets and the results of space-borne observations. The importance of this study is explained by the fact that even small variations of upper tropospheric water vapor can influence the magnitude of water vapor feedback (Brogniez and Pierrehumbert, 2006).

In the validation tasks, the importance of studying the land-sea contrast of atmospheric parameters rather than the values of these parameters over land and water separately arises from the fact that inconsistency of data can be detected more easily in this way. The vivid example of detecting inconsistency in data by means of looking at the land-sea contrast of atmospheric parameter is an artefact in ozone column measurements by the TOMS (Total Ozone Mapping Spectrometer) instrument (Cuevas, 2001). Persistent year-to-year differences in total ozone between continents and oceans were found in the mean global ozone data which were averaged in time. This feature has been named GHOST (Global Hidden Ozone Structures from TOMS). Part of these differences appeared to be caused by truncation of the lower tropospheric column due to the topography and by permanent differences in tropopause height distribution. The remaining part (66%) has been found to be an artefact of the retrieval algorithm: the effects of the presence of UV-absorbing aerosols might have been accounted for not correctly.

Previously, the measurements of cloud liquid water path (LWP) by the SEVIRI and AVHRR satellite instruments demonstrated the evidences of the systematic difference between the cloud amount and the LWP values over land and over water areas in Northern Europe (Karlsson, 2003; Kostsov et al., 2018b, 2019, 2021). The reason for the differences in spring and summer has been suggested by Karlsson (2003): the inflow of cold water from melting snow and ice is cooling the near-surface atmospheric layer over the water bodies. As a result, in contrast to the land surface, this layer over the water bodies becomes very stable preventing the formation of clouds. This mechanism, however, does not explain the existence of the LWP land-sea difference during cold season when both land and water surfaces are covered with snow and ice. So far not much attention was paid to the investigation of physical mechanisms which drive the LWP land-sea differences in Northern Europe.

New references:

Cuevas, E., Gil, M., Rodriguez, J., Navarro, M., and Hoinka, K.P.: Sea-land total ozone differences from TOMS: GHOST effect, Journal of Geophysical Research, 106 (D21), 27745-27755, https://doi.org/10.1029/2001JD900246, 2001.

Dong, B., Gregory, J.M., Sutton, R.T.: Understanding Land–Sea Warming Contrast in Response to Increasing Greenhouse Gases. Part I: Transient Adjustment, Journal of Climate, 22, 3079-3097, https://doi.org/10.1175/2009JCLI2652.1, 2009

Joshi, M., J. Gregory, M. Webb, D. Sexton, and T. Johns: Mechanisms for the land/sea warming contrast exhibited by simulations of climate change, Climate Dynamics, 30, 455–465, doi:10.1007/s00382-007-0306-1, 2008

Brogniez, H., and R. T. Pierrehumbert: Using microwave observations to assess large-scale control of free tropospheric water vapor in the midlatitudes, Geophys. Res. Lett., 33, L14801, doi:10.1029/2006GL026240, 2006

Chung, E.-S., B. J. Soden, B. J. Sohn, and J. Schmetz: An assessment of the diurnal variation of upper tropospheric humidity in reanalysis data sets, J. Geophys. Res. Atmos., 118, 3425–3430, doi:10.1002/jgrd.50345. 2013.

**Fig. 6(a) is not relevant since the scaling factor from this station was not used**

In the revised version of the paper we modified Fig. 6a by adding the values of scaling factor derived from the ceilometer observations at the location of the HATPRO radiometer. Though scaling factor from the station "St.Petersburg" is not used in calculations, it is used now for comparison with the ceilometer observations. We decided to keep Fig. 6a in order not to overload Fig. 6b with extra lines.

**Section 5: Validation of the off-zenith bias assessment using the FTIR instrument is difficult and could be removed, since no spatial information on clear sky conditions is obtained.**

We agree with this remark and removed this part of the text and Fig. 8.

**Fig. 7, 9, 10: the choice of showing monthly values while using a 10 day averaging period might not be ideal and information on the number of included cases is missing**

Definitely, there is a misunderstanding. Phrases with mentioned 10 day period are the following:

> *"It was found that for the described experiment the minimal time period for averaging is 10 days."*
> *(Line 248)*
> *"Averaging over a 10 day time period has been found to be sufficient for suppressing the random error due to FOV." (Line 372)*

These phrases indicate the **minimal** (or **sufficient**) period for averaging which has been obtained in the previous study (Kostsov et al., 2020). The reference to this previous study is present.

Phrases about averaging period in the current study are the following:

> *"The averaging of the individual measurements of the LWP land-sea difference was done over monthly periods." (Line 578)*
> *"The averaging of the individual measurements of the LWP land-sea difference was done over monthly periods." (Line 743)*

These phrases unambiguously declare that in the current study the period for averaging was one month.

As far as the remark about the number of included cases is concerned, we agree that this information can be useful and presented the new table with this information:

In Section 4:

> Table 3 shows number of measurements during each month which were selected for bias assessment. It should be emphasized that measurements were used regardless of illumination conditions.

In Section 5.1:

> Table 4 shows number of measurements during each month which were selected for assessment of the LWP land-sea contrast. It should be noted that the data in Tables 3 and 4 can not be compared since for assessment of the LWP contrast we selected measurements which were made under sun illumination conditions only (solar zenith angle less than 72°).

New tables:

**Table 3.** Number of measurements used for assessment of the LWP retrieval bias. Measurements during day and night were included.

| Month \ Year | 2013 | 2014 | 2015 | 2016 | 2017 | 2018 | 2019 | 2020 | 2021 |
|---|---|---|---|---|---|---|---|---|---|
| June | 289 | 186 | 414 | 93 | 328 | 308 | 241 | 127 | 64 |
| July | 118 | 281 | 35 | 162 | 176 | 210 | 227 | 76 | 225 |
| August | 87 | 269 | - | - | 248 | 379 | 517 | 339 | 397 |

**Table 4.** Number of measurements used for assessment of the LWP land-sea contrast. Measurements during day only were included (solar zenith angle less than 72°).

| Month \ Year | 2013 | 2014 | 2015 | 2016 | 2017 | 2018 | 2019 | 2020 | 2021 |
|---|---|---|---|---|---|---|---|---|---|
| June | 291 | 692 | 716 | 163 | 578 | 632 | 685 | 521 | 101 |
| July | 787 | 224 | 41 | 187 | 385 | 238 | 629 | 261 | 243 |
| August | 628 | 203 | - | - | 317 | 411 | 747 | 376 | 395 |

**Based on the comments above I don't recommend publishing this paper in its current form. The authors present similar conclusions about a land-sea contrast compared to a previous study using a regression retrieval for LWP and use highly uncertain assumptions for their physical retrieval approach and cloud scheme, while questions about the variability of the detected land-sea contrast still remain.**

As we noted above, the regression and physical approaches are used for processing microwave measurements for decades and in many cases the accuracies of these approaches are comparable. So, it is no wonder that the results for 2013 and 2014 provided by both algorithms are of the same range. However the present study makes a considerable step forward with respect to the previous study since many new aspects are analysed: cloud position and issues relevant to measurement geometry, atmospheric model for retrievals, cloud statistics, bias assessment, quality control procedure, validation issue.

We do not agree that the assumptions for physical retrieval approach are uncertain. The physical retrieval procedure for LWP and for other parameters from microwave measurements by HATPRO instrument is well tested in our previous studies where we compared physical and regression approach, ground-based and space-borne measurements of LWP over land and other studies. The physical procedure was only slightly modified for specific task of assessment of LWP contrast.

We agree with the esteemed referee that the some assumptions are uncertain which are used for scaling our results in order to compare them to the space-borne data on LWP contrast provided by SEVIRI instrument. We hope that our revisions concerning scaling factors help to assess at least the magnitude of this uncertainty.

And finally, concerning the conclusion of the referee that questions about the variability of the detected land-sea contrast still remain, we would like to note that the problem appeared to be rather complicated and requires further research, so we do not argue with this conclusion.

**Summary of main revisions:**

- The structure of the manuscript has been changed considerably: the application of scaling factors to the obtained LWP contrast values is now a separate part of study (Section 6) which refers only to comparisons with the satellite data (the conclusions have been changed accordingly). The table of contents now is the following:

*1 Introduction*
    *1.1 Background*
    *1.2 Motivation*
    *1.3 Novelty*

- The extensive uncertainty analysis has been provided.
- Ceilometer data were added for verification of scaling factors.
- The results have been validated using the approach of cross-validation by comparing the outputs of the physical and regression algorithms.
- The "standard" model of cloud altitude range 0.3-5.5 km was applied to data processing and the results were analysed
- Two extra elevation angles were considered and the results were analysed.

Vladimir Kostsov
(corresponding author)

13 May 2022

---

## Author Comment (AC2)

**The reply to the anonymous referee #2 (RC2)**

We are grateful to the referee for the very attentive reading of our manuscript and for many insightful remarks. We accept the criticism as very helpful. While preparing the revised version of our article, we took into account all comments made by the referee.

Below, the actual comments of the referee are given in **`bold courier font and blue colour`**. The text added to the revised version of the manuscript is marked by red colour.

*Notice: Since both anonymous referees made several similar remarks, our answers to these remarks which are given in both replies are identical.*

*Notice2: Numbering of figures and sections has been changed considerably in the revised version of the manuscript.*

**`Synthesis:`**
**`This study presents a method that can be applied to microwave radiometers operated close to a water surface to determine differences in cloud liquid water path over land and water. The results of the newly developed algorithm for the ground-based microwave radiometer is compared to satellite observations.`**

**`In the current form, I do not recommend the study to be published. There are several points that should be addressed, in particular a thorough uncertainty analysis. Please find my comments below.`**

**`General comments:`**

**`• The study lacks a thorough uncertainty analysis, from the brightness temperature uncertainties to the retrieval errors. You discuss the error sources in Chapter 3.2, but you do not provide any values for the different errors which would be crucial for interpreting the results. The LWP differences in your study on the order of much less than 10 g/m² are well within the error ranges (both bias and random errors). Please provide a detailed uncertainty analysis including error bars in Figures 6, 7, 9, 10, and 11 !`**

As a reply to this comment, we expanded the uncertainty analysis. Following the advice of the referee, we began with the brightness temperature uncertainties and LWP retrieval errors for zenith geometry in Section 3.2:

The input data for the retrievals are the values of brightness temperature of down-welling microwave radiation in 14 spectral channels of the HATPRO radiometer. In the so-called "humidity channels" which are located in the range 22.24-31.4 GHz, the random error of brightness temperature measurements are declared by the manufacturer of the instrument as 0.1 K. In the so-called "temperature channels" which are located in the range 51.26-58.0 GHz, the random error are declared to be 0.2 K. There is also a small systematic error which remains after calibration by liquid nitrogen. It can not be controlled but according to special studies does not exceed 0.5 K. The random errors of brightness temperature measurements have a direct influence on the estimation of the retrieval errors of all profiles of atmospheric parameters on the basis of the Fisher matrix calculations (Eq. 5). As noted above, the LWP retrieval errors are obtained with the help of Eq. 9. According to our earlier studies (Kostsov et al., 2018a), the bias of LWP retrievals for zenith geometry derived from cloud-free observations is very stable and constitutes 0.010 kg m$^{-2}$. The random error of the LWP retrieval has been estimated from cloud-free observations as 0.001 kg m$^{-2}$. The random errors of LWP retrieval derived from the error matrix calculations at the final iteration step of each retrieval are comparable to the estimations made on the basis of analysis of cloud-free periods and constitute in average 0.003-0.004 kg m$^{-2}$. We assume that the influence of the systematic brightness temperature error remaining after calibration is cancelled in the final LWP retrieval results by applying bias correction.

Physical retrievals imply calculations of brightness temperatures and kernels of the linearised radiative transfer equation for all spectral channels and elevation angles. Such calculations require accurate absorption models. Since start of operation of HATPRO at the observational station of St.Petersburg University, the absorption models for oxygen, water vapour and cloud liquid water were updated several times. At present the absorption model described by Rozenkranz (2017) is used, namely its version from 2019 (http://cetemps.aquila.infn.it/mwrnet/lblmrt_ns.html, last access 5 May 2022).

In the present study, we slightly modified the retrieval setup in order to provide equal sensitivity of zenith and off-zenith observations to LWP, see Section 3.3 below. As a result, the LWP retrieval errors estimated from Fisher matrix are somewhat higher than reported in earlier studies for routine zenith observations and constitute about 0.006-0.008 kg m$^{-2}$.

The assessment of the uncertainty of bias values in the revised version of the manuscript was done and described in Section 4:

The uncertainties of bias assessment were estimated in the following way:

- the three-step procedure described above was repeated with the value of the threshold 0.010 kg m$^{-2}$ at step 1 while other steps were kept the same;

- the differences between bias values obtained for thresholds 0.010 kg m$^{-2}$ and 0.005 kg m$^{-2}$ were attributed to the uncertainties of the bias assessment.

The uncertainties of bias assessment averaged over all months and years for zenith direction and two elevation angles 11.4° and 14.4° were found to be 0.002 kg m$^{-2}$, 0.001 kg m$^{-2}$ and 0.001 kg m$^{-2}$ respectively. These values are shown in Fig. 6 as error bars.

[Figure]

**Figure 6: The values of the LWP retrieval bias $b_i$ for zenith and off-zenith geometry derived for summer months in 2013-2021. Index i designates the elevation angle (1=90°, 2=14.4°, 3=11.4°). The uncertainties of bias assessment are shown as error bars in panel (a) only.**

The uncertainty of the values of the LWP land-sea contrast was obtained in the same way in Section 5.2:

The assessment of the uncertainty of the results was made similar to the assessment of the uncertainty of bias. Two values of the threshold for bias estimation were taken and the LWP contrast was derived for these two cases. The difference between the results was equal to 0.001 kg m$^{-2}$ if averaged over all months and years. It was attributed to the uncertainty of the obtained values of the LWP land-sea contrast and is shown in Fig. 8 in the form of error bars.

[Figure]

**Figure 8: The results of the retrieval of LWP land-sea contrast for summer months within the period 2013-2021. No scaling factors are applied. The uncertainties of the results are shown as error bars in panel (a) only.**

The uncertainty of the scaling factors is discussed below as a reply to the special remark which concerns scaling factors.

The uncertainty of the scaled multi-year averaged LWP contrast values has been estimated also. Fig. 14 now contains error bars. The text describing this figure has been slightly changed, see Section 6.3:

In Fig. 14 we demonstrate a comparison of the results of the assessment of the LWP land-sea contrast based on ground-based microwave measurements with the results of the satellite observations by the SEVIRI instrument. The satellite data were taken from the study by Kostsov et al. (2021). In the mentioned study, the time period 2011-2017 was analysed which partly overlaps with the time period considered in the present work (2013-2021). So, we compare the mean monthly values of the LWP contrast averaged over these overlapping 8-year and 9-year periods. The values $D_1$ and $D_2$ obtained from ground-based measurements have been averaged jointly. The following values of scaling factor $F_2$ were applied: 5.3±1.3 for June, 5.8±1.5 for July and 7.4±2.3 for August respectively (see Section 4 for derivation of these values). The uncertainty of the scaled LWP contrast values is shown as error bars in Fig. 14. It was estimated accounting both for the uncertainty of original data on the LWP contrast and the uncertainty of the scaling factor. The information on the uncertainty of the averaged SEVIRI data is not available.

Since the mean LWP contrasts for summer months are analysed, the so-called "August anomaly" should be mentioned which was revealed by the satellite observations and shows up as the practically total absence of the LWP contrast in August if compared to June and July. It should be emphasised that this "August anomaly" concerns the Gulf of Finland only; this effect is absent for neighboring large and small lakes (Kostsov et al., 2021). One can see that the ground-based measurements produce no evidence of this anomaly: the LWP land-sea contrast values for all summer months are almost the same. Moreover, the highest values are detected in August. For June and July, the results obtained from satellite observations are higher than the results obtained from ground-based measurements by 0.008-0.009 kg m$^{-2}$. The discrepancies are well within the limits determined by error bars for HATPRO data. To our opinion, the agreement between the space-borne and ground-based results for these two months can be estimated as very good. As it was mentioned above, the estimates of $F_2$ obtained in section 4 were the lower ones. In reality, $F_2$ can be larger and in this case the discrepancy between the ground-based and the satellite data for June and July will be smaller. As far as the comparison for August is concerned, the discrepancy goes far beyond the error limits, so the absence of the August anomaly effect has been verified quantitatively.

[Figure]

**Figure 14: The estimations of the LWP land-sea contrast in the region of Neva Bay of the Gulf of Finland for summer months. The results are shown which were derived from ground-based microwave measurements by the RPG-HATPRO instrument (scaled) and from space-borne observations by the SEVIRI instrument during multi year periods. The uncertainty of the scaled LWP contrast values from HATPRO is shown as error bars**

**• For your cloud retrieval you set up a so-called "cloud area of interest" which is between 1 and 4 km above ground. In general, there are many shallow clouds over the sea with both base and top below 1 km which would be completely neglected by this study. This is also confirmed by the cloud base observations used in the study (see line 481). It would be important to consider using lower elevation angles for the microwave radiometer to also catch the lower clouds. Otherwise, too many clouds are just not sampled for this study and a meaningful comparison with satellites becomes nearly impossible.**

Following the suggestion made by the referee, we considered using lower elevation angles and presented the results in Subsection 5.5:

**5.5 Test retrievals for elevation angles 19.2° and 8.4°**

It is interesting to investigate the dependence of the derived values of the LWP land-sea contrast on the elevation angle. It has been explained above that two elevation angles 11.4° and 14.4° have been selected as optimal for solving the task of detecting the LWP land-sea contrast for specific geometry of the experiment which is determined by the location of the instrument and the size of the water body. The lines of sight for these two angles intersect the area of interest over water body spanning vertically from 1 to 4 km. Despite the fact that there can be possible influence of the underlying surface on the results of microwave measurements when elevation angles are smaller than 10°, test retrievals were made for the elevation angle 8.4°. Also, test retrievals were made for the elevation angle 19.2° in order to obtain information which could be sufficient for derivation of a dependence on elevation angle. Both "extra" elevation angles fit well the atmospheric model compiled for the LWP contrast retrievals since the corresponding measurements still probe a significant amount of the designated area of interest (see Fig. 1).

Validation of the results for two extra elevation angles was made on the basis of comparison of the LWP contrast values derived by physical and regression algorithms. This comparison has shown that the discrepancy between the outputs of physical and regression algorithms for the angle 19.2° has the same magnitude as for the optimal angles 11.4° and 14.4°. For the angle 8.4°, the discrepancy is noticeably larger. This fact is the confirmation of the recommendation to avoid using elevation angles smaller than 10°. Nevertheless, we kept all results for analysis.

The values of the LWP land-sea contrast are plotted as a function of elevation angle in Fig. 11 for each summer month and each year separately. One can notice a clear and well pronounced dependence of the LWP contrast on elevation angle which is characteristic for all cases except two. Maximal contrasts are detected for the optimal elevation angles 11.4° and 14.4°. For "extra" angles, the contrasts are always lower (or sometimes equal to contrasts for optimal angles). Two exceptions refer to July 2014 and July 2016: in these cases the dependence on elevation angle is absent. The detected dependence can be explained by clear physical reasons. For large elevation angles, the portions of air, which are probed, are shifted up and towards the instrument. The liquid water content for high clouds usually is less than for the lower clouds, hence the contrasts should be smaller. Also, the contrast in the vicinity of the coastline is expected to be smaller. For small elevation angles, the air portion, which is probed, is shifted down and towards the opposite shore. The influence of clouds over the opposite shore can cause the decrease of mean values of the LWP contrast. One can see that in some cases the LWP contrast obtained for extra elevation angles is negative. These cases require special study, which should be based probably on the analysis of synoptic situations.

[Figure]

**Figure 11: Retrieved LWP land-sea contrast as a function of elevation angle for June, July and August in 2013-2021. Symbols correspond to four discrete values of elevation angle (19.2°, 14.4°, 11.4°, and 8.4°) and are connected by lines only for illustrative purpose.**

**● The setup of the study lacks some comparison to land areas for the low elevation angles. It would have been good to also perform scans to the other direction (i.e. south-west) to check if the algorithm performs well over land, too. With a HATPRO microwave radiometer these bi-lateral scans can be performed easily. This would have provided a reference area to see whether**

**the results between zenith and slant observations are in fact caused by the different surface conditions.**

We are grateful to the referee for proposing this idea. At present, we try to think over how to implement it. There is one obstacle. Our HATPRO radiometer is installed on top of a tower together with several other instruments. All instruments are powered from a stand which is located in the centre of a tower. If we simply turn HATPRO by 180°, the line of sight will meet this stand at low elevation angles. So we are now trying to solve the problem if it is possible to move HATPRO slightly aside without affecting the functionality of other instruments.

**• The assumptions for the physical algorithm are very rough and the information on the forward model is quite sparse. The a-priori profiles (page 6, lines 191-192) are not described. Where do you get these profiles from, especially also the information on cloud liquid water? Which assumptions are used concerning clouds? And what are the absorption models used in the study?**

We do not quite understand what assumptions are meant here. The physical algorithm is well tuned, tested and used for routine observations by the HATPRO radiometer at St.Petersburg University already for years. The reference to previous studies and information on retrieval accuracy are provided in the revised version in the beginning of subsection 3.2:

The input data for the retrievals are the values of brightness temperature of down-welling microwave radiation in 14 spectral channels of the HATPRO radiometer. In the so-called "humidity channels" which are located in the range 22.24-31.4 GHz, the random error of brightness temperature measurements are declared by the manufacturer of the instrument as 0.1 K. In the so-called "temperature channels" which are located in the range 51.26-58.0 GHz, the random error are declared to be 0.2 K. There is also a small systematic error which remains after calibration by liquid nitrogen. It can not be controlled but according to special studies does not exceed 0.5 K. The random errors of brightness temperature measurements have a direct influence on the estimation of the retrieval errors of all profiles of atmospheric parameters on the basis of the Fisher matrix calculations (Eq. 5). As noted above, the LWP retrieval errors are obtained with the help of Eq. 9. According to our earlier studies (Kostsov et al., 2018a), the bias of LWP retrievals for zenith geometry derived from cloud-free observations is very stable and constitutes 0.010 kg m$^{-2}$. The random error of the LWP retrieval has been estimated from cloud-free observations as 0.001 kg m$^{-2}$. The random errors of LWP retrieval derived from the error matrix calculations at the final iteration step of each retrieval are comparable to the estimations made on the basis of analysis of cloud-free periods and constitute in average 0.003-0.004 kg m$^{-2}$. We assume that the influence of the systematic brightness temperature error remaining after calibration is cancelled in the final LWP retrieval results by applying bias correction.

We added the information about a priori profiles in the end of subsection 3.1:

A priori profile of LWC is assumed as nearly zero profile (10$^{-8}$ kg m$^{-2}$) in the altitude interval (1-4 km) which simulates cloud-free conditions. In contrast, the variability of LWC is quite large which allows retrieving high range of LWP. For temperature and absolute humidity a priori profiles, we use the data from nearest radiosonde station which were averaged over several years of radiosonde launches.

We added the information about absorption models in subsection 3.2:

Physical retrievals imply calculations of brightness temperatures and kernels of the linearised radiative transfer equation for all spectral channels and elevation angles. Such calculations require accurate absorption models. Since start of operation of HATPRO at the observational station of St.Petersburg University, the absorption models for oxygen, water vapour and cloud liquid water were updated several times. At present the absorption model described by Rozenkranz (2017) is used, namely its version from 2019 (http://cetemps.aquila.infn.it/mwrnet/lblmrt_ns.html, last access 5 May 2022).

Rosenkranz, P. W.: Line-by-line microwave radiative transfer (non-scattering), Remote Sens. Code Library, https://doi.org/10.21982/M81013, 2017.

**• I'm not convinced by the definition of the scaling factors F1 and F2. You are assuming that the vertical distribution of clouds is the same over all the years. Other factors like air or water temperature anomalies, as well as**

the precipitation patterns (dry/wet months) will strongly influence the cloud
distribution of a special month which makes it very difficult to believe that
these scaling factors are stable for a special month. Did you see inter-
annual differences for the period of the cloud base height dataset? If so,
this would add uncertainty to the scaling factors. Furthermore, in Figure 6
it can be seen that the scaling factor depends very much on the location
where the cloud base height has been taken. Also, it can be seen that there
is quite a strong monthly variability. Do you have an explanation for this
behaviour?

We agree with the statement that scaling factors are not expected to be stable for a special month. And of course we agree that the precipitation patterns can influence the cloud distribution. We admit that Fig. 8 showing scaled monthly values for each year can be to some extent misleading because we apply average scaling factors to specific month of a specific year. Therefore we removed this figure and relevant discussion. We keep Fig. 10 since we believe that the results of comparison averaged over nine years are robust and valuable. In the revised version we emphasise that our comparison with satellite data which uses scaling factors is based on several strong assumptions and we estimate the uncertainty of LWP contrast values averaged over nine years.

In order to provide a kind of validation of scaling factors derived from statistics collected by human observers at meteorological stations we compared scaling factors from meteorological stations with the factors derived from ceilometer measurements. We also added the discussion of the uncertainty of scaling factors:

In order to provide a kind of validation of scaling factors derived from meteorological data we calculated scaling factors using the CBH measurements made with a ceilometer. The CHM 15k ceilometer was in operation at the observational site of St.Petersburg University in 2013, 2014 and the first half of 2015. It was installed on the metal tower on the roof of the building just nearby the HATPRO microwave radiometer. The full description of the CHM 15k ceilometer can be found on the web page of a manufacturer (https://www.lufft.com/products/cloud-height-snow-depth-sensors-288/ceilometer-chm-15k-nimbus-2300/, last access 6 May 2022). The CBH values were derived by the original data processing algorithm embedded in the instrument. For the purpose of comparison, the selection of CBH data for calculation of scaling factors was done exactly in the same way as the selection of data provided by human observers at meteorological stations.

Resulting monthly values of $F_1$ and $F_2$ for daytime are shown in Fig. 13. Since scaling factors obtained from the ceilometer observations appeared to be very similar to factors at the meteorological station St.Petersburg, they are shown in one plot Fig. 13a. The factors $F_1$ and $F_2$ from ceilometer observations are practically indistinguishable, therefore only the $F_2$ factor is shown in this plot. Comparison of $F_1$ and $F_2$ obtained from the St.Petersburg station records is given in Fig. 13a and demonstrates also very similar values for all months. Minimum monthly values of scaling factors from all data sources are observed in spring and summer, maximal values – in autumn and winter. In spring, summer and early autumn, the differences between the multi-year average values of $F_2$ from meteorological station and the values of $F_2$ for specific years are the smallest if compared to late autumn and winter. This fact is the strong indication that the cloud statistics are stable for these months. It is surprising that for June and July the results obtained from ceilometer measurements in 2013, 2014 and 2015 are nearly identical.

Comparison of scale factor $F_2$ derived from the records of all three meteorological stations is shown in Fig. 13b. One can see that the main feature of intra-annual variability is the same for all stations: minimal values in spring and maximal in late autumn. Apparently, this behavior is due to the predominance of the lowest clouds in late autumn, and of the medium and high clouds – in spring. But there are also noticeable differences. First, while $F_2$ for the St.Petersburg station is nearly constant from March to September, $F_2$ for the Kronstadt and Lomonosov stations increases during this period. From December to April, the values of $F_2$ for all three stations are very similar. From May to November, there is noticeable difference between the values obtained at St.Peterburg station and two other stations. For Kronstadt and Lomonosov, the scaling factor is approximately 1.5-2.5 times higher than for St.Petersburg.

Comparison of Fig. 13a with Fig. 13b leads to several principal conclusions. First, the scaling factors are essentially different in summer and autumn for so-called "continental" and "marine" locations. Second, the location of the HATPRO radiometer can be attributed to the category of "continental" locations. And third, while for "continental" locations there is some evidence of the stability of scaling factors for specific months, we can not

prove it for "marine" locations. As a result, the problem arises what scaling factor to choose and how to estimate the uncertainty of chosen factor. Since the main part of the line of sight of the radiometer passes over water body, it is reasonable to choose the scaling factor obtained at marine locations. The average scaling factors derived from the data obtained at the two meteorological stations Lomonosov and Kronstadt are 5.3, 5.8, and 7.4 for June, July, and August respectively. These values have been chosen for scaling the LWP land-sea contrast data (see Section 6 below). The half difference between these values and the correspondent values from the "continental" station St.Petersburg seems to be reasonable uncertainty estimate. So finally we have the following data for $F_2$: 5.3±1.3 for June, 5.8±1.5 for July and 7.4±2.3 for August.

[Figure]

Figure 13: (a) Scaling factors $F_1$ and $F_2$ derived from the data of meteorological observations of the cloud base height at the St.Petersburg station in 2011-2017 (average values) and from ceilometer observations in 2013, 2014 and 2015 at the location of the HATPRO radiometer. (b) Scaling factor $F_2$ derived from the data of meteorological observations of the cloud base height at three stations in the vicinity of the radiometer (see the legend) in 2011-2017 (average values).

- **Did you compare your LWP from the physical retrieval with statistical approaches? I would be interested to see whether there are differences, as you provided a new (physical) algorithm here.**

In the original version of the manuscript, we deliberately did not present the comparison of the results obtained by physical approach with the results obtained by regression approach because it was not the main focus of the work. However we agree with the referee that it is very important

comparison. Since the esteemed referee attracted attention to this issue, we clarified it in the revised version as follows:

**5.4 Cross-comparison of the LWP contrast values derived by the physical algorithm and by the regression algorithm as a means of validation of the obtained results**

We would like to discuss briefly the problem of choice of a retrieval algorithm (physical or regression) for the specific task which we solve in this study and the problem of validation of obtained results. The LWP contrast values for 2013 and 2014 obtained by the regression algorithm in the previous study (Kostsov et al., 2020 and Fig. 18 therein) are of the same range as ones produced by the physical approach in the present work. As one can see in Fig. 6, the years 2013 and 2014 are somewhat special: the LWP retrieval bias is minimal and comparable for zenith and off-zenith observations. It is no wonder that both algorithm work well. The situation is completely different for the years 2015-2021 when the LWP retrieval bias values for off-zenith observations are considerably larger. In order to compare the results produced by the two algorithms we made two tests. In the first test, we applied the regression algorithm to the HATPRO measurements which successfully passed quality control (convergence of the physical retrieval process and spectral residual check). In the second test we applied regression algorithm to all measurements (no quality control at all). It is important to emphasise that in both cases the bias of the LWP retrievals by regression method was estimated from measurements selected by the physical method. The outcome was quite demonstrative. In the first case, the results produced by the physical and regression algorithms were in good qualitative and quantitative agreement. They are demonstrated in Fig. 10, panels (a), (c), and (e). One can see the overall very good agreement with only two noticeable deviations of the results obtained by regression algorithm from the results of the physical algorithm (July 2016, August 2013). In the second test, the results obtained by the two algorithms were completely different and they are not shown. These tests clearly indicated the superiority of the physical algorithm in the problem of assessment of the LWP contrast. As it was noted in the beginning of Section 2, the identification of cloud-free periods of time and quality checks are impossible using the results obtained by the regression method. Therefore, it is necessary to emphasise once again that for the data set of HATPRO measurements, which we process in the present study, the application of the regression algorithm only would have produced wrong results.

Comparison of the results obtained by different algorithms can be a valuable means of cross-validation. A vivid demonstration of such cross-validation has been presented in the study by Kostsov et al. (2018a). In this study a special case of ground-based measurements by HATPRO radiometer after a rain event was considered. After rain, the radome of microwave instrument is wet for some time. During this period of time measurements are erroneous. Kostsov et al. (2018a) have shown that the discrepancy between the outputs of regression and physical algorithms is maximal during a rain and becomes smaller and smaller after a rain event while water evaporates from the radome. This effect clearly demonstrates that controlling the discrepancy between the outputs of regression and physical algorithms can be a means of validation of the final results. This statement refers not only to an "after a rain" period but has a general character and can be applied to different sophisticated situations and problems like the problem of detecting the LWP land-sea contrast. That is why, one can consider the good agreement of LWP contrast values produced by regression and physical algorithms as a successful validation of the obtained results.

In order to get one more confirmation of the validity of the obtained results, we processed the HATPRO data applying the "standard" atmospheric model which is used in routine zenith microwave observations and is characterized by the cloud altitude range 0.3-5.5 km. The data processing procedure was the same as for the model with cloud altitude range 1-4 km, with two exceptions:
   - the threshold for bias assessment was taken as 0.015 kg m-2 since the "standard" model has larger a priori uncertainty of cloud liquid water and hence larger bias;
   - the retrieval setup was not specially tuned to provide equal sensitivity of zenith and off-zenith measurements to atmospheric parameters, the retrieval setup for the model with cloud altitude range 1-4 km was used without any modifications.

The results of the derivation of the LWP land-sea contrast with the "standard" model are presented in Fig. 10, panels (b), (d), and (f). One can see that the discrepancies between the outputs of physical and regression algorithms are noticeably larger than for the "1-4 km" model. Also, the differences between $D_1$ and $D_2$ values for both algorithms are very much larger than ones obtained for the "1-4 km" model. These two facts indicate that in case of "standard" model the results are very much less self consistent than for the "1-4 km" model. Comparison of left and right panels in Fig. 10 leads to the conclusion that the modification of "standard" model for the specific task of detecting the LWP land-sea contrast was the correct decision.

[Figure]

**Figure 10: Comparison of the LWP land-sea contrast values ($D_1$ and $D_2$) obtained by the physical algorithm and by the regression algorithm for June (a, b), July (c, d) and August (e, f) in the period 2013-2021 and for two cloud altitude range models: 1-4 km (a, c, e) and 0.3-5.5 km (b, d, f).**

● **For the LWP bias correction (Figure 7), why are there sometimes considerable differences between zenith and off-zenith observations, and sometimes not (e.g. 2020 in June vs. July?). Are you sure that your bias correction algorithm is working properly? What might be the reason for these differences?**

We agree with this remark and we added the discussion of the variability of bias values in Section 4:

We tried to find the reasons for the variability of bias values for off-zenith geometry. The primary guess was the influence of different synoptic situations, namely the situations with different integrated water vapour (IWV). However, this guess was not confirmed. As shown in Fig. 7a, there is no noticeable correlation between the monthly mean values of bias $b_2$ and monthly mean values of IWV during the period 2013-2021. The important notice should be made: we calculated mean IWV values exactly for the time periods which were selected for bias assessment. In Fig. 7a one can notice large inter-annual variability of IWV for all summer months. Lowest mean IWV values are detected in June, and the highest values are detected in July. We made an attempt to find a correlation between monthly mean values of bias $b_2$ and monthly mean values of temperature at the ground level during the period 2013-2021, but no noticeable correlation has been found. As a result, the conclusion was made that the influence of synoptic situation on LWP bias was not detected.

Another idea about the reason for off-zenith bias variations was related to possible horizontal inhomogeneities of temperature in the vicinity of the radiometer. It has been already noted that the radiometer is

installed on the roof of the building which is heated by solar irradiance. The building is about 200 m long and the line-of-sight passes directly over it. Fig. 7b shows the values of the LWP retrieval bias $b_2$ for summer months in 2013-2021 as a function of mean value of the temperature horizontal difference $\Delta T$ at 100 m altitude near the radiometer for these months. This temperature difference was derived from the temperature profiles which were retrieved using zenith and off-zenith geometry. Of course, one should keep in mind that the spatial resolution is very poor, but nevertheless this temperature difference can be used either an indicator of horizontal temperature gradients or an indicator of some effects which interfere in the retrieval process. The assessment of correlations between $b_2$ and $\Delta T$ using the Fisher criterion for small number of samples has shown that correlations in July and August are statistically significant. The correlation is also statistically significant if the entire summer period is analysed without division by months. So we accepted the hypothesis that the reason for the variability of LWP bias for off-zenith observations can be a temperature horizontal inhomogeneity in the close vicinity of the radiometer. Probable mechanism could be the following: under conditions of considerable temperature inhomogeneity, the temperature error propagates into the LWP error and the LWP bias increases.

[Figure]

**Figure 7: (a) The values of the LWP retrieval bias $b_2$ for summer months in 2013-2021 as a function of mean value of IWV for these months. (b) The values of the LWP retrieval bias $b_2$ for summer months in 2013-2021 as a function of mean value of the temperature horizontal difference at 100 m altitude near the radiometer for these months. Dashed lines demonstrate linear fits. The values of correlation coefficient are given as *r*.**

**Minor comments:**
**- Page 2, lines 40-46: Too much detailed information**

We completely removed the presentation of scanning radiometers. Instead, we added brief discussion of the importance of studying land-sea contrasts of atmospheric parameters.

**- Page 5, line 125: I would use the term "approach" instead of "algorithm"**

The term "algorithm" was replaced by the term "approach".

**- Page 5, line 145: What do you mean by "model form"?**

In order to make it clear, we changed the sentence in the following way:

> The applicability of the physical method to the problem of the LWP and IWV retrieval by two-channel radiometers implies that the a priori profiles of pressure, temperature and humidity are available from external data sources and the cloud liquid water content (LWC) profile is taken from some statistical or numerical cloud model.

**- Page 12, lines 315-316: Why do you think that the LWP contrast is the same for cloud scenes with different cloud base heights! I don't think that is a valid assumption!**

We believe that in the considered specific case this assumption is pretty good due to two reasons. First, the difference in CBH for low and medium clouds is less than 700 m, so the LWP contrasts are not expected to differ considerably. And second, the frequency of occurrence for high clouds is much less than of low and medium clouds, therefore the contribution of error caused by different LWP contrasts for high clouds will be small. In the revised text we wrote this explanation:

However, to achieve this goal, one additional assumption should be made: the mean values of the LWP contrast for cloud scenes with different CBH are assumed to be similar:

$$D_{medium} = D_{low} = D_{high} \qquad (24)$$

We believe that in our case with the quite large cloud altitude range of interest (1-4 km) this assumption is pretty good due to two reasons. First, the difference in CBH for low and medium clouds is less than 700 m, so the LWP contrasts are not expected to differ considerably. And second, the frequency of occurrence for high clouds is much less than of low and medium clouds, therefore the contribution of error caused by different LWP contrast for high clouds will be small.

**- Page 17, line 452: I don't understand what you want to say here. Do you mean that the sensitivity is instrument specific? If so, then please write it that way.**

No, we just mean that the manipulations with the assignment of measurement error values can not increase the sensitivity because we can not use the error value smaller than the quantity which is specific for our instrument and is declared by the manufacturer. In order to avoid confusion we decided to remove the unclear sentence.

**Summary of main revisions:**

- The structure of the manuscript has been changed considerably: the application of scaling factors to the obtained LWP contrast values is now a separate part of study (Section 6) which refers only to comparisons with the satellite data (the conclusions have been changed accordingly). The table of contents now is the following:

- The extensive uncertainty analysis has been provided.
- Ceilometer data were added for verification of scaling factors.
- The results have been validated using the approach of cross-validation by comparing the outputs of the physical and regression algorithms.
- The "standard" model of cloud altitude range 0.3-5.5 km was applied to data processing and the results were analysed
- Two extra elevation angles were considered and the results were analysed.

Vladimir Kostsov
(corresponding author)

13 May 2022